# Initialization of a global glacier model based on present-day glacier geometry and past climate information: an ensemble approach

Julia Eis[1], Fabien Maussion[2], and Ben Marzeion[1,3]

[1]Institute of Geography, University of Bremen, Bremen, Germany
[2]Department of Atmospheric and Cryospheric Sciences, University of Innsbruck, Innsbruck, Austria
[3]MARUM - Center for Marine Environmental Sciences, University of Bremen, Bremen, Germany

**Correspondence:** J. Eis (jeis@uni-bremen.de)

**Abstract.** To provide estimates of past glacier mass changes over the course of the 20[th] century, an adequate initial state is required. However, empirical evidence about past glacier states at regional or global scale is largely incomplete, both spatially and temporally, calling for the use of automated numerical methods. This study presents a new way to initialize the Open Global Glacier Model from past climate information and present-day glacier states. We use synthetic experiments to show that even with these perfectly known but incomplete boundary conditions, the problem of model initialization is an ill-posed inverse problem leading to non-unique solutions, and propose an ensemble approach as a way forward. The method works as follows: we generate a large set of physically plausible glacier candidates for a given year in the past (e.g. 1850 in the Alps), all of which are then modelled forward to the date of the observed glacier outline and evaluated by comparing the results of the forward runs to the present-day states. We test the approach on 2660 Alpine glaciers and determine error estimates of the method from the synthetic experiments. The results show that the solution is often non-unique, as many of the reconstructed initial states converge towards the observed state in the year of observation. We find that the median state of the best 5 percent of all acceptable states is a reasonable best estimate. The accuracy of the method depends on the type of the considered observation for the evaluation (glacier length, area, or geometry). Trying to find past states from only present-day length instead of the full geometry leads to a sharp increase in uncertainty. Our study thus also provides quantitative information on how well the reconstructed initial glacier states are constrained through the limited information available to us. We analyse which glacier characteristics influence the reconstructability of a glacier, and discuss ways to develop the method further for real-world applications.

## 1 Introduction

Glaciers contributed significantly to past sea-level rise (SLR; e.g. Gregory et al., 2013; Slangen et al., 2017a; Cazenave and et al., 2018; Wouters et al., 2019; Zemp et al., 2019) and they will continue to be a major contributor in the 21[th] century (e.g. Church et al., 2013; Slangen et al., 2017b; Hock et al., 2019). A large fraction of this contribution will be caused by

the ongoing adjustment of glaciers to previous climate change (Marzeion et al., 2014, 2018). Reconstructions of past glacier mass change are therefore not only necessary to determine the budget of past sea-level change (Gregory et al., 2013) and to increase the confidence in projections (by allowing to quantify the agreement with observations, Marzeion et al., 2015), they also enable us to quantify the pattern of the ongoing adjustment of glaciers to present-day climate. Estimates of global glacier
mass change are based on in-situ measurements in mass and length changes (e.g. Zemp et al., 2015; Leclercq et al., 2011), on remote sensing techniques (e.g. Gardner et al., 2013; Jacob et al., 2012; Bamber et al., 2018; Wouters et al., 2019), or on mass balance modelling driven by climate observations (Marzeion et al., 2012, 2015). Since observations of temperature, and to a smaller degree, precipitation, are more ubiquitous (e.g. Harris et al., 2014) than glacier observations (WGMS, 2018), reconstructions of glacier change produced by forcing a glacier model with climate observations have the potential to increase
the understanding of past glacier behaviour. Finally, reconstructing glacier change based on climate model output allows to test the skill of climate models (Goosse et al., 2018).

A number of global glacier models were developed in the past (e.g., Radić and Hock, 2011, 2014; Giesen and Oerlemans, 2012, 2013; Marzeion et al., 2012, 2014; Huss and Hock, 2015; Maussion et al., 2019). The more recent and complex of these models (e.g. Huss and Hock, 2015; Maussion et al., 2019) require Digital Elevation Models (DEMs) and outlines from the
Randolph Glacier Inventory (RGI; Pfeffer et al., 2014) to derive the initial surface hypsometry. Hence, their starting date of a glacier evolution simulation depends on the recording date of the DEM and the outline, which typically do not coincide with one another, nor with the required starting date of a projection. The model of Huss and Hock (2015) indicates a high sensitivity to the initial ice volume. Similarly, Maussion et al. (2019) remark that great uncertainties, especially on local and regional scales, derive from unknown initial conditions.

Despite the importance of glacier contribution to past sea-level rise, so far only one model was able to provide estimates of glacier mass changes over the course of the entire 20[th] century on the global scale (Marzeion et al., 2012). All other global modelling studies limit their application to the recent past and future projections. The reconstruction by Marzeion et al. (2012) was possible because of the highly parameterized representation of ice dynamics and glacier geometry change, applying a Volume-Area-Time scaling to translate mass change into surface area and elevation range changes. Based on this approach, it
was sufficient to iteratively optimize one variable (glacier size in the year of interest, e.g., 1850) such that when run forward to the year of the observed glacier outline, the modelled glacier area agreed with the observed glacier area.

An increase of model complexity impedes the process as more and more variables are required for initialization. Flowline models require input data along the coordinates of the flowline (e.g. bed topography, surface elevations and/or widths) and thus more complex initialization methods are needed. For example, van Pelt et al. (2013) developed an iterative inverse method to
reconstruct distributed bedrock topography and simultaneously initialize an ice flow model. Zekollari et al. (2019) added an ice flow model to Huss and Hock (2015), which required an automated initialization for glaciers in 1990 (prior to the glacier inventory date) to avoid spin-up issues and so that the reconstructed initial states fit the glacier geometry at the inventory date after being modelled forward. By choosing a decade long initialization, they avoid problems of non-uniqueness (as we discuss below), but raise the question of how arbitrarily this date can be chosen. Similar approaches exists for the initialization of ice
sheet models, where most work focuses on estimating the present-day state of ice sheets in order to make accurate projection of

future ice sheet change (e.g. Heimbach and Bugnion, 2009; Lee et al., 2015; Mosbeux et al., 2016). Goelzer et al. (2018) divide the existing initialization approaches into 3 methods: spin-up, assimilation of velocity, and assimilation of surface elevation. Spin-up procedures are typically used for long-term and palaeoclimate simulations, the required spin-up time is unknown and can be relative long. Additionally, the reconstruction cannot be expected to represent effects from internal climate variability correctly. The data assimilation approaches typically determine model parameters (e.g. basal parameters like basal friction or bedrock topography) that reduce the mismatch between observed and modelled velocities or surface elevations.

In this study, we aim to identify fundamental limitations that narrow the reconstruction of past glacier states from present day geometries, under the assumption of perfectly known boundary conditions and a perfect glacier model. Specific research questions are:

- To which degree does the past evolution of a glacier constrain its present day geometry?

- How much information does the present day glacier geometry contain about its past states?

- Is it possible to reconstruct past glacier states from the partial information available to us?

- How far can we go back in time to have an initial geometry that still determines the present-day glacier geometry?

- Which glacier attributes influence the answers to the questions above?

To this aim, we present a new method estimating past glacier states and apply it to synthetic numerical experiments, and we show the obstacles that need to be overcome before applying our method to real-world problems. After introducing the relevant features of the Open Global Glacier Model (OGGM; Maussion et al., 2019) in Sect. 2.1, we describe the design of the synthetic experiments in Sect. 2.3. The synthetic framework serves to test the skill of our approach in a surrogate model world where everything is known, and allows to apply data denial experiments to address the questions listed above. The initialization method is presented in Sect. 2.4. The developed method consists of 3 steps: generation of plausible glacier states, identification of glacier state candidates, and their evaluation based on the misfit between the modelled and the observed geometry at the year of the observation. We applied our approach to 2660 Alpine glaciers and present the results for the reconstructed initial states in the year 1850 in Sect. 4.1. The influence of the considered type of observation (e.g. glacier length, area or geometry) is shown in Sect. 4.2 and a statistical analysis of glacier attributes that influence the reconstructability of a glacier is presented in Sect. 4.3. Finally, we summarize the results and discuss the limitations of the method and its applicability to real-case studies, as well as needed and possible future developments in Sect. 6.

## 2 Methods

### 2.1 The Open Global Glacier Model

The Open Global Glacier Model (OGGM; Maussion et al., 2019) is an open source numerical framework that allows the modelling of past and future changes of any glacier in the world. Starting with a glacier outline, provided by the Randolph

Glacier Inventory (RGIv6.0; Pfeffer et al., 2014), a suitable surface DEM is automatically downloaded and interpolated to a local grid. The size of the local grid is given by a border parameter, which is the number of grid points outside the glacier boundaries. We choose a border value of 200 grid points to ensure that also large glacier states can be generated. The resolution of the map topography $dx$ depends on the size of the glacier ($dx = a\sqrt{S}$, with $a = 14\text{m km}^{-1}$ and $S$ the area of the glacier in

km$^2$) and is constrained to $10\text{ m} \leq \text{dx} \leq 200\text{ m}$. After the preprocessing, glacier centerlines are computed using a geometrical routing algorithm (adapted from Kienholz et al., 2014). They are then considered as glacier flowlines, and grid points are generated using a fixed, equidistant grid spacing, which is twice that of the underlying 2D map topography. Surface elevations along the flowline coordinates are then obtained from the underlying topography file and glacier section widths are computed by intersecting the normal of the flowline to the boundaries of the glacier. By making assumptions about the shape of the bed

(parabolic, rectangular or a mix of both), OGGM estimates the ice thickness with a mass-conservation approach (Farinotti et al., 2009, 2017, 2019). Information on bed topography at each grid point results from the calculated ice thickness and the surface elevation. From this, the glacier length, area, and volume can be determined. These values depend strongly on the surface topography and are based on the (often wrong) assumption that the recording date of DEM and that of the outline coincide. The dynamical flowline model of OGGM can then be used to determine the evolution of the glacier under any given

climate forcing by solving the shallow ice approximation along the flowlines.

The mass balance is computed at each grid point using climate data (monthly temperature and precipitation). Climate data can be used from different sources, including gridded observations or reanalyses for past climate, projections for future climate, or randomized climate time series. The purpose of forcing the mass balance model with randomized climate is to easily produce a great number of realistic climate forcings representative of a given time period, characterized by a center year $y_0$ and a window

size $h$ (typically 31 years). All climate years $\in [y_0 - \frac{h-1}{2}, y_0 + \frac{h-1}{2}]$ are then shuffled infinitely in the next step. Additionally, it is possible to set a temperature bias $\beta$, which shifts all values of the temperature series towards warmer or colder climates. Identically to the study of Maussion et al. (2019), we only calibrate the mass-balance model while the creep parameter $A$ and the sliding parameter $f_s$ are the same for each glacier and set to their default values ($A = 2.4 \times 10^{-24}\text{s}^{-1}\text{Pa}^{-3}, \text{f}_\text{s} = 0$, no lateral drag). The following mass balance related parameter values were used in this study: $p_f = 1.75$, $T_{Melt} = -1.5°\text{C}$,

$T_{Liquid} = 2.0°\text{C}$ and $\Gamma = -6.5\text{Kkm}^{-1}$. This parameter set was determined with a cross-validation done with the HISTALP data set and tested for the 41 Alpine glaciers with more than 5 years of mass-balance observation. For more details concerning the glacier model (e.g. the mass-balance calibration or sensitivities to the dynamical parameters of the model) please refer to Maussion et al. (2019) and http://docs.oggm.org.

## 2.2 Problem description

Here, we define a *glacier state* (hereinafter referred as *state*) as follows:

**Definition 1.** *Let $m \in \mathbb{N}$ be the total number of grid points of all flowlines of a glacier. Then $s_t = (z_t, w_t, b)$ is a **glacier state** at time $t$, with surface elevation $z_t \in \mathbb{R}_+^m$, widths $w_t \in \mathbb{R}_+^m$, and bed topography $b \in \mathbb{R}_+^m$. The set $\mathcal{S}_{t_i} = \{s_t | t = t_i\}$ contains all physically plausible glacier states at time $t_i$.*

The construction of an initial state is an inverse problem and can be defined in opposition to the direct problem. The *direct problem* corresponds to a forward model run: given an initial state $s_{t_0} \in \mathcal{S}_{t_0}$ at time $t_0$, the state $s_t \in \mathcal{S}_t$ at time $t > t_0$ can be computed by:

$$s_t = G_{\text{past}}(s_{t_0}) \tag{1}$$

where $G_{\text{past}} : \mathcal{S}_{t_0} \to \mathcal{S}_t$ is an operator representing the equations of OGGM, using known climate time series as boundary condition.

For *inverse problems*, the solution is known by direct observations: $s_{t_e} = s_{t_e}^{obs}$, whereas the desired initial state $s_{t_0}$ is unknown. The inverse problem consists of finding the initial state $s_{t_0} \in \mathcal{S}_{t_0}$, such that the forward modelled solution at time $t_e$ fits the observations from the same year $t_e$:

$$s_{t_0} = G_{\text{past}}^{-1}(s_{t_e}^{obs}) \tag{2}$$

Unfortunately, we do not have an explicit formulation for $G_{\text{past}}^{-1}$ in our case. A backwards reconstruction is impeded by the non-linear interaction between glacier geometry, ice flow and mass balance. Optimization methods can be used to solve inverse problems. To this end, we introduce a minimization problem such that the forward modelled state is as close as possible to the observation:

$$\min_{s_{t_0} \in \mathcal{S}_{t_0}} j(s_{t_0}) \tag{3}$$

with

$$j(s_{t_0}) := \frac{1}{m} \parallel s_{t_e}^{obs} - \underbrace{G_{\text{past}}(s_{t_0})}_{s_{t_e}} \parallel_2^2 = \frac{1}{m} \left( \sum_{i=0}^{m} \left( (z_{t_e}^{obs})_i - (z_{t_e})_i \right)^2 + \left( (w_{t_e}^{obs})_i - (w_{t_e})_i \right)^2 \right) \tag{4}$$

This function calculates the averaged difference in surface elevation and width between the observed and forward modelled glacier state. Differences in bed topography can be neglected, as we assume the bed topography to remain the same over the inspected time period.

In many cases, however, OGGM forward integrations of different initial states result in very similar states at time $t_e$. This implies that there exist many local minima of the function $j(s_{t_0})$. As uncertainties of the model can safely be assumed to be larger than the differences between those states at time $t_e$, it is impossible to identify the global minimum of $j(s_{t_0})$. I.e., the solution of our inverse problem is non-unique.

The objective of our approach is therefore to identify the set $\mathcal{S}_{t_0}^{\epsilon}$ of all states, which correspond to the observed state $s_{t_e}^{obs}$ within a given uncertainty $\epsilon$ after being modelled forward. We call this condition *acceptance criterion*:

$$J(s_{t_0}) := \frac{j(s_{t_0})}{\epsilon} < 1 \tag{5}$$

The function $J(s_{t_0})$ is called in in the following *fitness function*. Assuming a vertical error of 5 m in $x$ and an horizontal error of 10 m in $w$, we propose to set $\epsilon = (5\text{m})^2 + (10\text{m})^2 = 125\text{m}^2$. These numbers can be changed easily, and in a real-world

application should be based on the vertical uncertainty of the reconstructed ice thickness and the horizontal uncertainty of the used outline. All states $s_{t_0} \in \mathcal{S}_{t_0}^{125}$ that have a fitness value smaller than one are called *acceptable states*. The first expectation would be that the glacier candidate with the smallest fitness value is also the best solution. However, due to uncertainties that derive from the model integration itself, this is not always the case. As an alternative, we determine the 5th quantile of all states

in $\mathcal{S}_{t_0}^{\epsilon}$. This set contains the best solutions of all acceptable states referred to their fitness values. We choose the median state as a representative of this set and compare the state with the minimal fitness value and the median state in Sect 4.1.

## 2.3   Synthetic experiments

### 2.3.1   Design

We create a time series of glacier states which range from the target year of initialization $t_0$ (e.g. 1850) to the present date $t_e$

(see Fig. 1 for an example). These are the glacier states that we aim to reconstruct with our initialization method, using only partial information (here the "observed" state at present day). This type of experiment is sometimes called "inverse crime" in the inverse problems literature (e.g. Colton and Kress, 1992; Henderson and Subbarao, 2017), and we explain their rationale below. To generate them, we apply a random climate scenario (window size $h = 31$ years and center year $y_0 = t_0 = 1850$) and run the model 600 years forward (see Fig. 1c). The temperature bias is set to $\beta = -1K$ to ensure that a relatively large 1850

glacier state is created (as expected for most real glaciers at the end of the Little Ice Age). The resulting state is defined to be the synthetic experiment state in year $t_0$ (see Fig. 1a). We model this state forward, applying the past climate time series from $t_0$ until $t_e$ (here: 2000) (see Fig. 1d) and obtain the *observed* state of the synthetic experiment (see Fig. 1b). Thanks to the initial temperature bias of $\beta = -1K$, these synthetic states in $t_e$ are very close to the real observed states in 2000 on average (total area difference for the Alps of about 1%, but individual glaciers can vary) and the total synthetic glacierized area in 1850

fits well to an estimate of Zemp et al. (2006) (see appendix A for more details). We call the states derived from the synthetic experiment $s_t^{exp}$.

### 2.3.2   Rationale

These synthetic states therefore provide a realistic setting with a strong advantage over actual observations: they are perfectly

known. As stated in the introduction, reconstructing past glacier states is a complex inverse problem, which accuracy will depend on (i) the uncertainties in the boundary conditions (climate, glacier bed, etc.), (ii) the uncertainties in the glacier model itself, and (iii) a theoretical lower bound (termed "reconstructability" in this study) tied to the characteristics of the glacier itself (slope, size, the past climate, etc.). The main objective of the synthetic experiments is to separate these issues from one another, and to focus on point (iii) only. This allows us to isolate and understand the limitations and errors of the developed

method itself, as opposed to uncertainties that derive from unknown boundary conditions and model parameters. They also allow us to realize data denial experiments and detect which kind of observations are necessary to reduce the uncertainties of

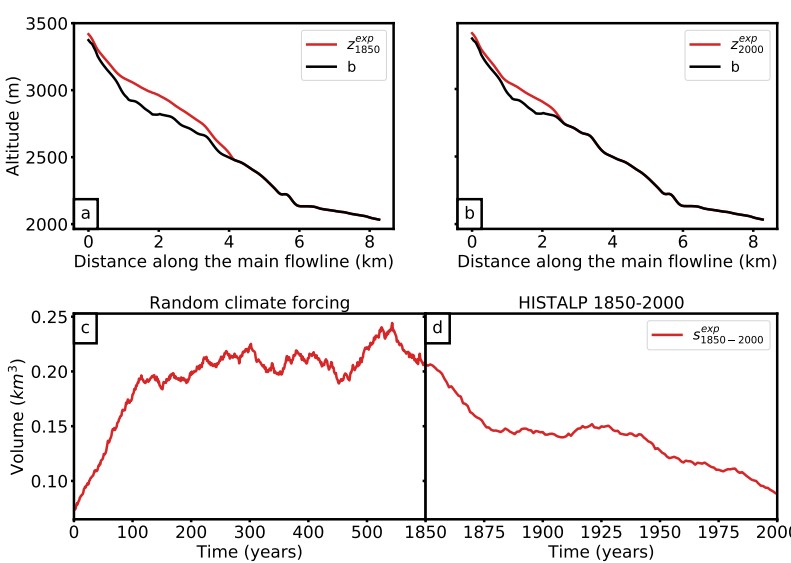

**Figure 1.** Illustration of the generation of the synthetic experiment with the example of Guslarferner (Oetztal, Austria). **a:** glacier thickness along the main flowline at $t_0 = 1850$ and **b:** $t_e = 2000$. The black line indicates the bed rock and the red line the ice surface of the synthetic experiment. **c:** generation of $s_{1850}^{exp}$, which is the state at t=600 (the end of the trajectory) and **d:** the volume of the glacier states $s_t^{exp}$ from 1850 to 2000. Note that the synthetic year 2000 glacier does not necessarily correspond to the "true" year 2000 glacier.

our reconstruction (Sect. 4.2), and to determine which glacier characteristics affect the reconstructability of a glacier (Sect. 4.3).

## 2.4 Reconstruction of initial glacier states

Our initialization method consists of three main steps: generation of a set of physically plausible glacier states $\mathcal{S}_{t_0}$, identification of glacier candidates $s_{t_0} \in \mathcal{S}_{t_0}$, and their evaluation based on the fitness function $J(s_{t_0})$ (see Sec. 2.2).

### 2.4.1 Generation of potential glacier candidates

In a first step, we generate a set of different, physically plausible states from which we will pool our "candidates" (Fig. 2a). For this purpose, we utilize a random mass balance model with a window size of $h = 31$ years and the center year $y_0 = t_0$ to create different climate conditions. Obviously, we do not use the same permutation as for the creation of the synthetic experiments (Sect. 2.3). This procedure generates a climate representative of a given time period with an interannual variability uncorrelated to that of the original period. For each random climate a different way of permutation is used. This ensures that all generated climate time series differ from each other, but at the same time all represent the climate conditions around $t_0$ (and an associated temperature bias $\beta$). The infinite permutation is necessary to obtain a time series that is long enough for the glaciers to reach

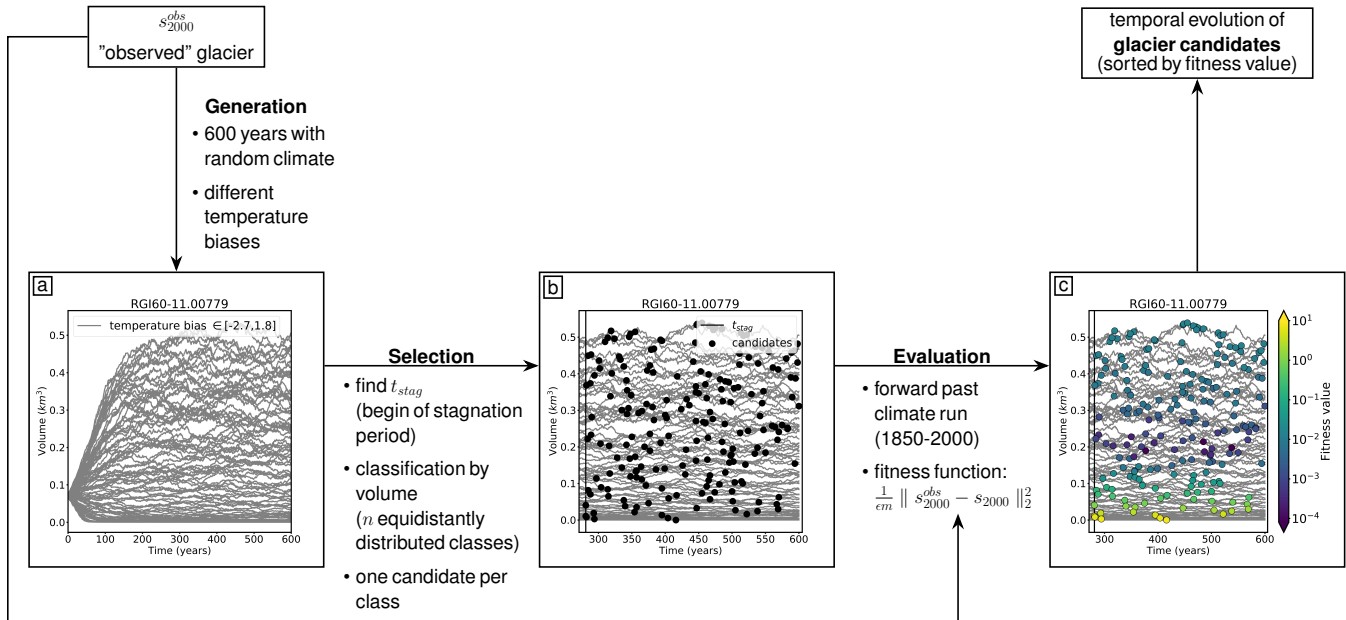

**Figure 2.** Workflow of the candidates generation, selection and ranking method, using Guslarferner (Oetztal, Austria) as an example. **a:** generation of potential glacier candidates. The grey lines indicate the glacier volume evolution for a set of different random climate scenarios over 600 years each. The temperature biases vary between -2.7 and 1.8 K. **b:** selection of candidates. The black vertical line indicates $t_{stag}$ and the black points show 200 candidates. **c:** glacier candidates colored by their fitness value. Violet marks candidates with a small misfit, whereas yellow marks states that don't meet the acceptance criterion (Eq. 5).

an equilibrium (while maintaining the impact of interannual climate variability) with the forcing climate (here 600 years). To create a large set of states, we additionally vary the temperature bias $\beta$. Glaciers respond differently to changes in climate and thus the required temperature biases vary from glacier to glacier and have to be inferred. We start with temperature biases $\beta \in [-2, 2]$ K. If $\beta = 2$ K is not large enough to result in a present day glacier with zero ice thickness, higher values will be used. If $\beta = -2$ K is not small enough to result in a glacier that reaches the boundary of the local grid (200 grid points outside of the glacier outline), smaller values will be used.

### 2.4.2 Selection of candidates

Figure 2a shows the evolution of the volume of the generated glacier states over time. In the first years, the time series clearly diverge (mostly caused by the temperature bias $\beta$), but after a certain time all time series begin to fluctuate around an equilibrium value. We refer to the period of fluctuations around the assumed equilibrium as the *stagnation period*. During the stagnation period the glacier volume does not increase or decrease strongly in comparison to the total volume change since the beginning of the simulation. We define $t_{stag}$ as the point in time where all trajectories have reached this stagnation period

and choose the upper ten volume trajectories, corresponding to the lowest temperature biases, to determine $t_{stag}$. To this end, we smooth each of the ten curves with a 10 years rolling-window and calculate the time point of their first maximum. $t_{stag}$ is defined as the latest of all previously determined time points (see Fig. 2b).

Defining $t_{stag}$ is necessary, because we determine initial glacier states at $t_0 = 1850$ and the searched glaciers are assumed to
be in equilibrium with the climate around 1850. Hence, each state that fluctuate around an equilibrium value is a potential glacier state candidate (in the following referred as candidate). This holds true for all states $s_t$ with $t > t_{stag}$. Depending on $t_{stag}$ and the number of successfully completed random climate runs $n_r$ (number of grey lines in Fig. 2) the sample size is $n_r(600 - t_{stag})$ (glacier states are stored yearly). The sample size is sufficiently large for all cases, e.g. in the case of the Guslarferner (Fig. 2) the sample contains approx. 44.500 members. In order to avoid testing very similar states, we classify all
states by their volume and select one candidate per class. We choose $n$ equidistantly and approximately uniform distributed classes, where $n$ (default: $n = 200$) is the number of candidates to evaluate in step three.

### 2.4.3   Evaluation

The last step evaluates all previously selected candidates. Each candidate is used as initial condition for a forward run, using observed past climate time series, e.g. from $t_0 = 1850$ until $t_e = 2000$. All runs use the same model parameter set, except for
the initial condition and exactly the same climate time series (e.g. no temperature bias $\beta$ is applied for the past climate runs). Afterwards, we compare the resulting modelled states $s_{t_e}$ with an observed state $s_{t_e}^{obs}$ (here taken from the synthetic experiment) by applying the fitness function $J(s_{t_0})$ (Eq. 5). This function calculates the averaged difference between the glacier geometries at the grid points, more specifically between the surface elevations $z_{t_e}$ and the widths $w_{t_e}$, of the observed and the modelled glacier. In Fig. 2c the candidates are colored by their fitness value.

## 3   Test site and Input Data

We tested our approach on Alpine glaciers. The glacier outlines are taken from the Randolph Glacier Inventory (RGI v6.0, region 11; Pfeffer et al., 2014). We use topographical data from the Shuttle Radar Topography Mission (SRTM) 90m Digital Elevation Database v4.1 (Jarvis et al., 2008). The SRTM aquisition date (2000) matches well that of the RGI (2003 for most glaciers). The climate dataset we use for this approach is the HISTALP database (Auer et al., 2007, http://www.zamg.ac.at/
histalp). The temperature time series covers the period 1780 to 2014 and the precipitation time series 1801 to 2014. Both data sets are available on a regular grid of 5 minutes resolution (approx. 9.3 km in the Alps).

We generate synthetic experiments (see Sect. 2.3) for all glaciers in the Alps, and determine their glacier states in 1850 if the area of the observed synthetic state $s_{2000}^{exp}$ is larger than 0.01 km$^2$. This value is consistent with the minimum-area threshold of the RGI. The condition is satisfied for 2660 synthetic experiments of the 3927 glaciers included in the Randolph Glacier
Inventory in Central Europe (region 11).

## 4 Results

Here we show the results for two example glaciers in 1850 as well as an error analysis for all tested glaciers (Sect. 4.1), the influence of the choice of the fitness function on the quality of our results (Sect. 4.2), and a statistical analysis of glacier attributes (including glacier response time) that influence the reconstructability of a glacier (Sect. 4.3).

### 4.1 Initial glacier states in 1850

Following the method described in Sect. 2.4, we determine reconstructed initial glacier states in $t_0 = 1850$. Figures 3 and 4 show the results of the Guslarferner, as an example glacier with a large set of accepted candidates. A second case with a more narrow set of acceptable states, the Hintereisferner, is shown in Fig. 5 and 6. More examples can be found in the supplementary material.

Especially the result of the Guslarferner shows clearly that the determination of past states is not unique (see Fig. 3). Multiple initial states (violet and blue colored) merge to the observed state in the year of observation. The fitness values, which means the averaged difference between the forward modelled states and the observation at $t_e = 2000$, are extremely small for most candidates. The fitness values of all candidates range from $1.08 \times 10^{-6}$ to 7.98. Only 16 of the 200 candidates have a fitness value higher than one and thus do not fulfill the acceptance criterion (Eq. 5); for these states, the glacier in 1850 is too small to reach the volume of the observed glacier within 150 years. Also Fig.4 illustrates the diversity of the different acceptable solutions (grey, shadowed area). The length of all states in $S_{1850}^{125}$ varies between 0.98 km and 8.1 km. The acceptance criterion in this example is not strong enough to provide any information about the searched state in $t_0 = 1850$, as any of the candidates would lead to an acceptable result. Figure 4 also shows the 5th percentile of all acceptable states (blue, shadowed area). This set contains the 5 % best solutions, based on the fitness value. All candidates of the 5th percentile are in close proximity to the synthetic experiment. The range of fitness values of all candidates of the 5th percentile is $[1.08 \times 10^{-6}, 7.95 \times 10^{-5}]$ and the length of the states in 1850 only varies from about 3.6 km to 5.3 km. All these candidates match the synthetic experiment in $t_e = 2000$ very well and converge to the synthetic experiment by 1900 at the latest, which can be seen in Fig. 4c. As a representative of this set, we choose the median state of the 5th percentile of $S_t^{125}$ (in the following referred as $s_t^{med}$). Figure 4a shows that the surface elevation of $s_{1850}^{med}$ in 1850 corresponds very well to the synthetic experiment, whereas the state with the minimum fitness value (in the following referred as $s_t^{min}$) mismatches the synthetic experiment at the tongue of the glacier. Regarding the volumes, $s_t^{med}$ match exactly the volume of the synthetic experiment in 1850, whereas the volume of $s_t^{min}$ differs by 0.4 km$^3$.

In the case of the Hintereisferner (see Fig. 5) the fitness values of most candidates are large compared to the ones of the Guslarferner. Only a few candidates have extremely small fitness values and the past state is thus much more narrowly confined. The different states need more time to adapt to the climate conditions and therefore they do not converge as quickly to one state. As a result, the differences between the forward modelled states and the observed one in 2000 are larger. The fitness values of all candidates range between $2.8 \times 10^{-5}$ and 43.

36 candidates fulfill the acceptance criterion (Eq. 5). Figure 6 shows that the acceptance criterion in this case confines the result

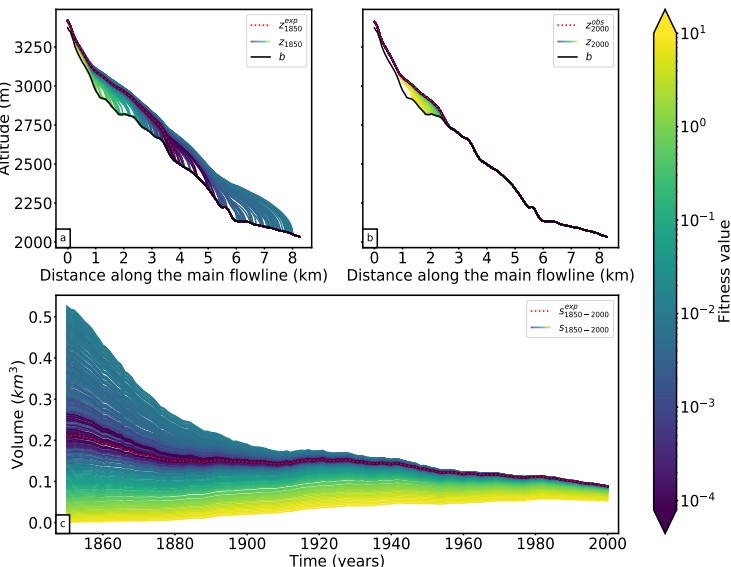

**Figure 3.** Results for the Guslarferner (Oetztal, Austria). Top: Cross-sections along the main flowline in **a:** 1850 and **b:** 2000. Black line indicates the bed rock, the red, dotted line the surface elevation from the synthetic experiment, and the remaining lines the modelled ice surfaces of all candidates, colored by their fitness value. The synthetic experiment state in 2000 has a length of 2.7 km, an area of 1.71 km$^2$, a volume of 0.09 km$^3$ and a mean thickness of 62.8 m. Bottom: Volume changes from 1850-2000, colored by their fitness values.

better than in the case of the Guslarferner. The length of all glaciers in $S_{1850}^{125}$ range from 8.4 km to 12.3 km. In this case the 5th quantile of $S_t^{125}$ is again in close proximity to the synthetic experiment and all candidates of the 5th quantile have extremely small fitness values (between 2.8 $\times 10^{-5}$ and 5.4 $\times 10^{-4}$). The length of the candidates of the 5th quantile in 1850 only varies from 9.1 km to 10.3 km and is thus more precise than in the Guslarferner example. In this example, all candidates of the

5   5th percentile converge no later than 1920 to the state of the synthetic experiment. Here $s_t^{min}$ matches the surface elevation of the synthetic experiment in 1850, as well as the volume trajectory over time, slightly better than $s_t^{med}$, but the volume differences to the synthetic experiments in 1850 are also very small in this example (0.004 $km^3$ for $s_t^{min}$ and -0.08 $km^3$ for $s_t^{med}$ ).

For both examples we were able to show that our method is able to recover the state in $t_0 = 1850$ of the synthetic experiment

10   by only using information about the observed state of the synthetic experiment in $t_e = 2000$ and combining it with information about the past climate evolution. $s_t^{med}$, as well as $s_t^{min}$ match the synthetic experiment in $t_0 = 1850$ extremely well. In the following, we provide an error analysis including all glaciers in the Alps on which we applied our method to. For each of the 2660 glaciers we have calculated the absolute volume error to the synthetic experiment:

$$e_{abs}^{med/min}(t) = v^{med/min}(t) - v^{exp}(t), \tag{6}$$

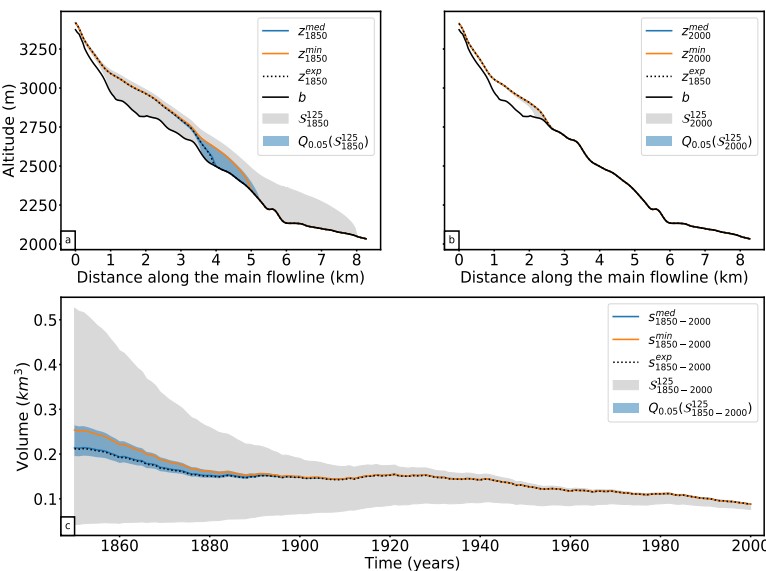

**Figure 4.** Results for the Guslarferner (Oetztal, Austria). Top: Cross-sections along the main flowline in **a:** 1850 and **b:** 2000. Bottom: Volume changes from 1850-2000. The grey shaded area indicates the range of all solutions with a fitness value smaller than one ($\mathcal{S}_t^{125}$). The blue shaded area shows the range of the 5th quantile of $\mathcal{S}_t^{125}$, the blue line $s_t^{med}$, and the orange line $s_t^{min}$.

where $v^{exp}(t)$ is the volume of the synthetic experiment in year $t$ and $v^{med/min}(t)$ is the volume of $s_t^{med}$ or $s_t^{min}$ in the same year $t$. Figure 7a shows the absolute volume errors in $km^3$ for $s_t^{med}$, as well as for $s_t^{min}$. Whereas the absolute volume errors in 1850 vary widely from approx. -1.1 $km^3$ to 2.9 $km^3$, they reduce rapidly within 60 years. In 1910, the errors range from approx. -0.25 $km^3$ to 0.17 $km^3$. The range of errors in the first 60 years is largely influenced by a few single outliers.

5    Differences between $s_t^{min}$ and $s_t^{med}$ are small. Figure 7b shows the median and the range of the 5th-95th percentile of $e_{abs}^{med}$ and $e_{abs}^{min}$ over time, indicating the robustness of our method. The median of $e_{abs}$ of both analyzed states is very small; 0.00028 $km^3$ for $s_t^{min}$ and 0.00076 $km^3$ for $s_t^{med}$ in 1850. The improvement with time can also be seen here: the median of $e_{abs}^{med}(1910)$ is of the order of $10^{-5}$ $km^3$ and that of $e_{abs}^{min}(1910)$ of the order of $10^{-6}$ $km^3$.

As our test site contains large and small sized glaciers, we also evaluate relative errors (in %):

$$e_{rel}^{med/min}(t) = \frac{e_{abs}^{med/min}(t)}{v^{exp}(t)} * 100 \tag{7}$$

Figure 8a shows the histogram of the relative errors in 1850, whereas the evolution from 1850-2000 of the median and the 5th-95th percentile of the relative errors are shown in Fig. 8b. The median of the relative volume errors in 1850 is -0.97 % for $s_t^{med}$ and -2.69 % for $s_t^{min}$. The 95th percentile value of $e_{rel}^{med}$ is 70 %. With 48% the value of $e_{rel}^{min}$ is smaller for the $s_t^{min}$. Whereas $s_t^{min}$ have in 1850 a slightly smaller 5th-95th percentile range than $s_t^{med}$, the median error of $s_t^{med}$ is slightly smaller

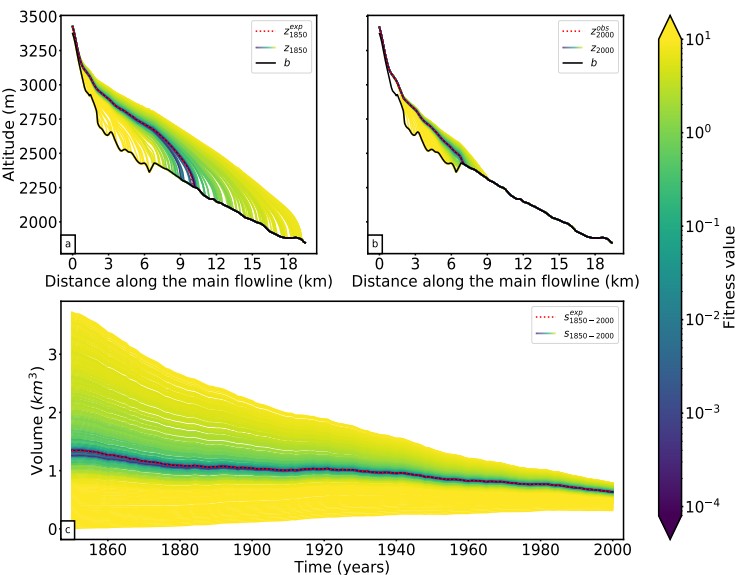

**Figure 5.** Results for the Hintereisferner (Oetztal, Austria). Top: Cross-sections along the main flowline in **a:** 1850 and **b:** 2000. Black line indicates the bed rock, the red dotted line the surface elevation from the synthetic experiment ,and the remaining lines the modelled ice surfaces of all candidates, colored by their fitness value. The synthetic experiment state in 2000 has a length of 7.3 km, an area of 7.76 km$^2$, a volume of 0.63 km$^3$ and a mean thickness of 105.5 m Bottom: Volume changes from 1850-2000, colored by their fitness values. All violet and blue glacier states merge to the observed glacier in 2000.

than the one of $s_t^{min}$.

Both states fit well the synthetic experiment. In many cases, $s_t^{med}$ is equal to $s_t^{min}$, but for some glaciers either $s_t^{min}$ or $s_t^{med}$ have a clearly better performance. In all cases, the uncertainties quickly reduce after around 1900 to 1930.

Figure 8a also shows that the error distribution is skewed and our method has a slight tendency to underestimate the glacier
5    volume. Although 64% of the relative errors have a negative sign, a few large positive outliers influence the mean error and shift it to a positive value of 16% (in 1850) for the minimum states or 23% (in 1850) in case of the median states.

## 4.2    Impact of the fitness function

For the evaluation of the glacier candidates we used a fitness function based on differences in the geometry of the glacier (see Eq. 5). In this section we want to test the influence of limited information on glacier geometry on the reconstructability of past
10   glacier states. Thus, we additionally evaluate the candidates by only using information about the glacier area or length.

For the glacier area based evaluation, we used the following fitness function:

$$J_A(A_{t_e}) = (A_{t_e}^{obs} - A_{t_e})^2 \tag{8}$$

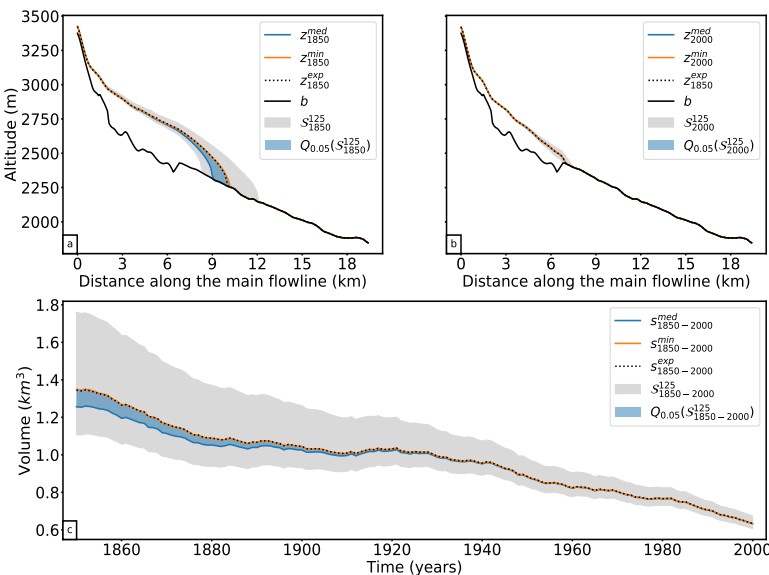

**Figure 6.** Results for the Hintereisferner (Oetztal, Austria). Top: Cross-sections along the main flowline in **a:** 1850 and **b:** 2000. Bottom: Volume changes from 1850-2000. The grey shaded area indicates the range of all solutions with a fitness value smaller than one ($\mathcal{S}_{1850}^{125}$). The blue shaded area shows the range of the 5th quantile of $\mathcal{S}_t^{125}$, the blue line shows $s_t^{med}$, and the orange line $s_t^{min}$.

where $A_{t_e}$ is the glacier area at time $t_e$. The fitness function that takes only information about the glacier length $l(t_e)$ at time $t_e$ into account is similar:

$$J_l(l_{t_e}) = (l_{t_e}^{obs} - l_{t_e})^2 \tag{9}$$

For each glacier in our test site, we evaluate the 200 candidates also with the fitness functions $J_A$ and $J_l$. For each evaluation
5  method (geometry, area and length based), we determine the state with the minimal fitness function [1] and calculate the relative volume error to the synthetic experiment.

Figure 9 shows the relative errors of all three evaluation methods. Figure 8a shows the distribution of the relative errors of the 5th-95th percentile in 1850, whereas the the evolution from 1850-2000 of the median and the 5th-95th percentile of the relative errors are shown in Fig. 8b. The more information is taken into account for the evaluation, the smaller are the errors.
10  The greatest uncertainties are associated with using the glacier length based fitness function (Eq. 9), whereas the differences between the area based evaluation (Eq. 8) and the geometry based evaluation (Eq. 5) are small. While the median errors in 1850 of the geometry and the area based evaluation are close (-2.69% for the geometry and -2.83 % for the area approach), the

---

[1] Instead, it is also possible to choose $s_t^{med}$ for the uncertainty analyses, but this would require acceptance criteria for the fitness functions $J_A$ and $J_l$, which would have influence on the state. For simplification, we choose the state with the minimal fitness function.

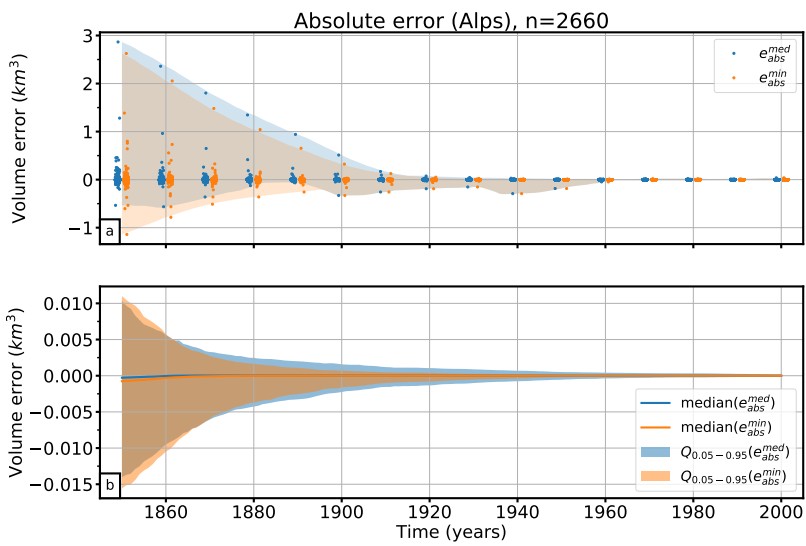

**Figure 7.** Absolute volume errors in km$^3$ over time of $s_t^{min}$ and $s_t^{med}$ of all tested glaciers. **a:** The blue points mark all individual errors $e_{abs}^{med}$ of $s_t^{med}$ and the orange points mark all individual errors $e_{abs}^{min}$ of $s_t^{min}$. The blue, shadowed area shows the total range of the errors $e_{abs}^{med}$ and the orange, shadowed area the total range of errors $e_{abs}^{min}$ over time. **b:** The shadowed areas show the 5th and 95th percentile ($Q_{0.05-0.95}$) of the absolute volume errors $e_{abs}^{med}$ (blue) and $e_{abs}^{min}$ (orange ) over time, as well as the median of $e_{abs}^{med}$ (blue line) and $e_{abs}^{min}$ (orange line).

median error in 1850 of the glacier length evaluation has with 107% the worst performance. This also applies for the values of the 95th precentile; 95% of the tested cases have in 1850 a relative volume error smaller than 1043%, if the length based fitness function is used for the evaluation. In contrast to the other two evaluations, this approach overestimates the volume. For the area based evaluation 95% of the tested glaciers have an error smaller than 90% and for the geometry based fitness function the error is smaller than 49%. This shows that the advantage using the geometry instead of the glacier area to evaluate the candidates is not very high; both evaluations shows a very good performance. Especially if the states are modelled forward (e.g. to 1900), both approaches perform well. However, it is not advisable to use the glacier length based evaluation.

## 4.3 Reconstructability

The examples from Sect. 4.1 as well as in the supplementary material indicate a high variation of the number of viable reconstructed candidates between glaciers. This number can range from a few viable solutions in a well defined range to many solutions without any constraints (all tested candidates have the same fitness value). In other words, some glaciers can be reconstructed easily, and some cannot.

We define a new measure of reconstructability $r$, where we set the volume range of the 5th percentile in relation to the volume

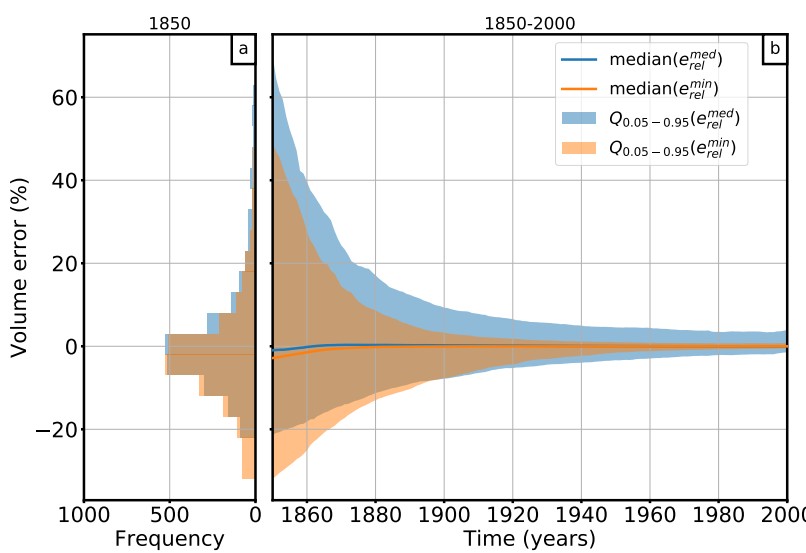

**Figure 8.** Relative volume errors of $s_t^{med}$ (blue colored) and $s_t^{min}$ (orange colored). Figure **a:** shows a histogram of all errors in the 5th-95th percentile in year 1850 and **b:** the evolution of the relative errors from 1850-2000. The line indicate the median error and the shadowed area the 5th-95th percentile range $Q_{0.05-0.95}$.

range of all acceptable states of the glacier:

$$r = 1 - \frac{\text{range}(Q_{0.05})}{\text{range}(S^{125})} \tag{10}$$

For a glacier with a unique solution, this measure is equal to one. If all accepted candidates have exactly the same fitness value, the measure will be zero (this occurs if all candidates converge to exactly the same state before the year 2000). Thus, a small

5 measure represents a glacier with low reconstructability and a measure close to one imply a higher reconstructability of the glacier. For example, $r$ is equal to 0.857 for Hintereisferner, and 0.879 for Guslarferner. The similarity of the two values can be explained by the similar proportion of the range of the 5th percentile to the range of all acceptable states in both cases (see Fig. 3 and 6). A histogram of the reconstructability values of all 2660 tested glaciers in the Alps is shown in Figure 10a. The distribution is bimodal and slightly skewed towards a high reconstructability. Values in the middle range are rare.

What glacier characteristics will influence this reconstructability? The working hypothesis is that it is likely to be associated with the concept of glacier "response time" (here formulated qualitatively). Glaciers with a short response time tend to be less sensitive to initial conditions, and will "forget" their initial state after a short period of time. This will probably lead to low reconstructability values. Inversely, glaciers with a long response time will be easier to reconstruct.

15 To test this hypothesis, we used the e-folding approach (as defined in Oerlemans, 1997, 2001) and calculated the time response to a step function. To this end, we first run the 1850 state of the synthetic experiment glacier into an equilibrium state by using

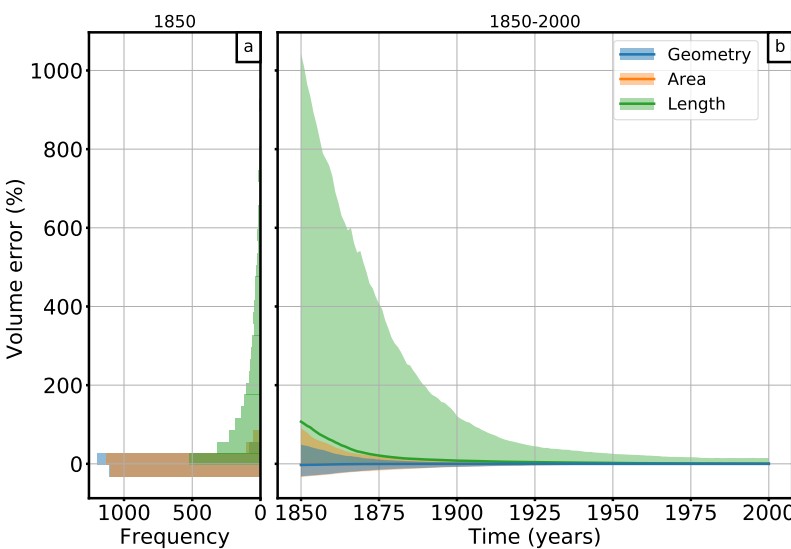

**Figure 9.** Relative volume errors of $s_t^{min}$ derived from different fitness functions based on the geometry (blue), the glacier area (orange) and glacier length (green). Figure **a:** shows a histogram of all relative errors in the 5th-95th percentile in year 1850 and **b:** the evolution of the relative errors from 1850-2000. The line shows the median error and the shadowed area the 5th-95th percentile range.

a constant climate (mean climate of the years 1835-1865, temperature bias = -1 K). We choose the same settings that were used for the generation of the synthetic experiments in order to obtain an equilibrium state $s_{eq1}$ close to our synthetic experiment in 1850. Next, we apply to $s_{eq1}$ a constant climate obtained by the mean climate of the years 1850-1880 using no temperature bias and receive the corresponding equilibrium state $s_{eq2}$ (i.e. a step change of 1 K). We calculate the e-folding time of these

5 two states for each glacier, but exclude the glaciers where the volume of $s_{eq2}$ reaches zero (which was the case for approx. 500 glaciers out of 2660[2]).

The scatter plot in Fig. 10b indicates a relation between the reconstructability measure and response time. The variance of the response time increases for reconstructability values close to one. Dependencies with the reconstructability could also be detected for the position of the equilibrium line altitude (ELA) (Fig. 10c), the mean surface slope in 2000 (Fig. 10d) and the

10 mean surface slope in 2000 of the last third of the glacier (Fig. 10e).

Furthermore, we calculated correlations of both reconstructability and response time with the following variables: glacier length (in 2000), area (in 2000), volume (in 2000), equilibrium line altitude (ELA) in 2000, equilibrium line altitude change from 1850-2000, mean surface slope (in 2000), and mean surface slope over the lowest third of the glacier (in 2000) (Fig. 11). The variable explaining reconstructability best is the glacier response time (correlation: 0.54). Both values correlate with the

15 same glacier characteristics. Against a common misunderstanding, glacier length, area and volume do not correlate well with

---

[2]We also tested a step of 0.5 K, leading to a larger sample size but no significant change to the correlation analysis and our results. Thus, we kept the 1 K step change here.

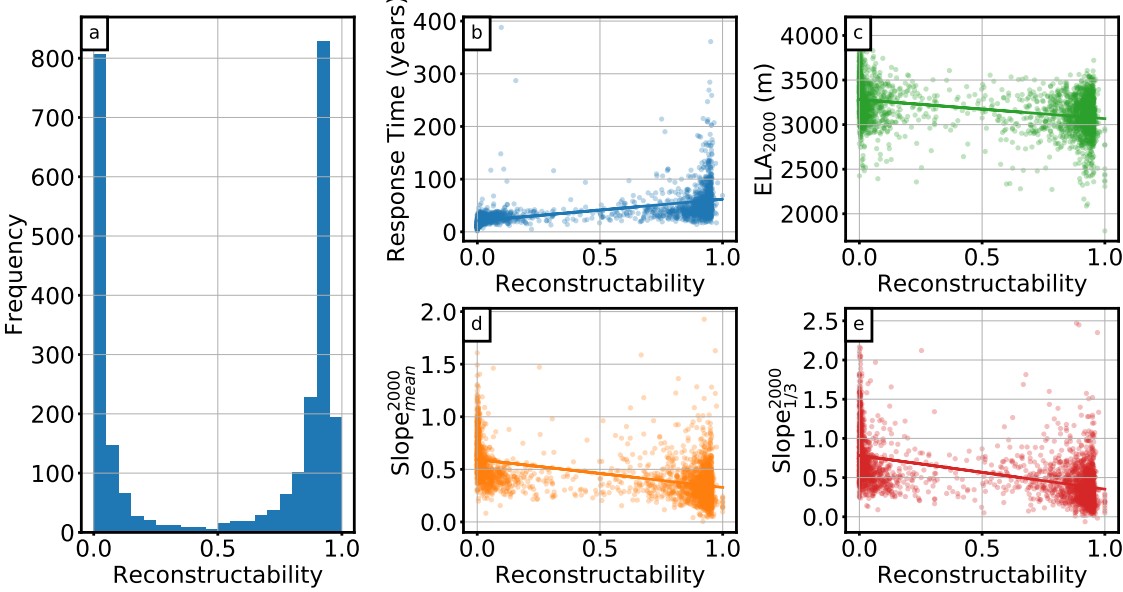

**Figure 10.** Reconstructability measure. **a:** Histogram of the reconstructability measure of all 2660 glaciers. **b-e:** Scatter plots with linear regression. The x-axis always shows the reconstructability measure. The y-axis shows **b:** the e-folding response time (n=2149), **c:** the equilibrium line atitude in 2000 (n=2660), **d:** the mean surface slope in 2000 (n=2660) and **e:** the mean surface slope of the last third of the glacier in 2000 (n=2660).

the reconstructability measure nor with the response time. The variable having the main influence is slope: generally, the larger the slope, the lower the reconstructability measure or the response time of a glacier. These findings coincide with results from Lüthi (2009), Zekollari and Huybrechts (2015) and Bach et al. (2018), who concluded that response times depend more on the steepness of the surface than on the glacier size. The correlation of the mean surface slope can be further increased by taking the lowest third of the glacier. Besides that, the position of the ELA in 2000 also influences the reconstructability, whereas the ELA change from 1850 to 2000 only plays a minor role.

Taken alone, these correlation values remain quite low and do not provide enough predictive power to create a statistical model of "reconstructability". However, they provide a good indication about which factors should be taken into account for future applications.

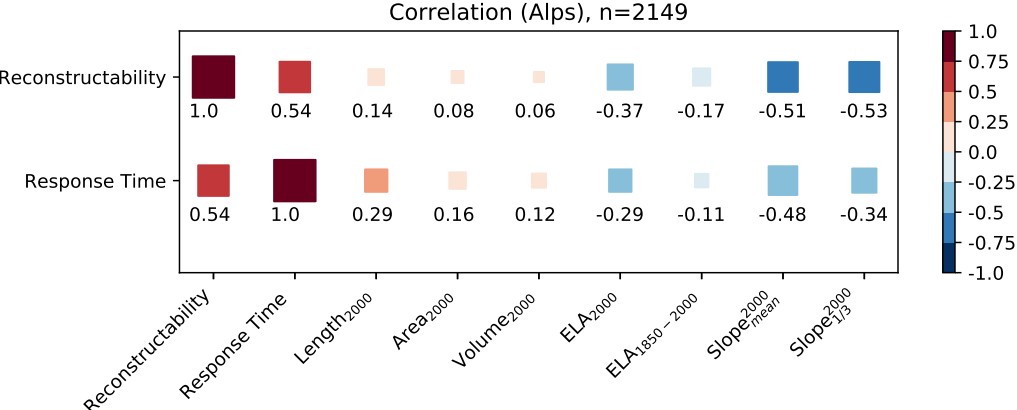

**Figure 11.** Correlation of the reconstructability measure and the e-folding response time to various glacier characteristics. Correlation values are represented by the square color and size.

## 5 Hardware requirements and performance

For this study we used a small cluster comprising two nodes with two 14-core CPUs each, resulting in 112 parallel threads (two threads per core). Our method requires to run hundreds of dynamical model runs for each single glacier and, as described in Maussion et al. (2019), the dynamical runs are the most expensive computations. The size of the glacier and the required time stepping to ensure a numerical stability strongly influences the required computation time. The computation time needed to apply our initialization procedure to one glacier varies from 30 seconds to 26 minutes. In total, initializing the 2660 glaciers in 1850 takes about 3.75 days on our small cluster.

## 6 Discussion and conclusions

In this study, a new method to initialize past glacier states is presented and applied to synthetic experiments. Assuming a perfectly known world allows us to identify the errors of our method alone and to separate them from uncertainties in observations and errors introduced by model approximations, a task impossible to realize in real-world applications. However, the synthetic experiments do not allow external validation, e.g against past outlines derived from moraines, historical maps or remote sensing (such as provided by GLIMS, Raup et al., 2007). Model uncertainties will have to be accounted for, and will have to be compared to and added to the theoretical lower bound discussed in this study. Similarly, our results do not provide information about actual past glacier mass change. Since in our synthetic experiments glaciers states in 2000 may be different from the real ones, the modelled initial glacier states in 1850 do not correspond to reality either. The past states determined in this study only can serve to verify the functionality of the developed method.

Our results have shown that the solutions are not unique. Multiple candidates match the observation in $t_e = 2000$, sometimes with a large spread. This raises interesting questions about the use of past glacier change information to reconstruct climate variations, which we don't address here. In the context of model initialization, this non-uniqueness is impeding the reconstruction. We evaluated the candidates with a fitness function based on averaged geometry differences between the forward modelled and the observed state in $t_e = 2000$. The threshold value $\epsilon = 125$ m$^2$ was derived by assuming a typical error of 5 m in surface elevations and 10 m in glacier width, but how these values should be chosen depends on the specific glacier setting. Especially in cases where many of the candidate states have extremely small fitness values, a more strict acceptance criterion can help to narrow the results. On the other hand, an $\epsilon$ that is too strict could lead to none of the candidates fulfilling the criterion.

Due to uncertainties that derive from the model integration, the glacier state with the minimal fitness value is not always close to the synthetic experiment. As a more robust alternative, we propose to use $s_t^{med}$, the median state of the 5th percentile of all acceptable states as the best estimate. In Sect. 4.1 we compared the errors of both approaches. The median error of $s_t^{med}$ is slightly smaller than that of $s_t^{min}$ and the total range of absolute errors is smaller for $s_t^{med}$ in 1850, too. Modelling the reconstructed initial states forward in time approximately 60 years leads to a rapid reduction of the error, and $s_t^{min}$ perform a bit better than $s_t^{med}$. By making use of the knowledge about the past climate, the number of candidates at later stages are through this forward run more constrained than by initializing them directly at a later time (see appendix B for a more detailed description of the inverted approach at different times).

By comparing different fitness functions for the candidate evaluation, we showed that using limited information only (glacier area and glacier length) lead to an increase of the errors in 1850. This indicates what kind of observation is needed to be able to reconstruct past glacier states from today's state. The differences between the geometry based evaluation and the area based evaluation are small, but the differences to the length based evaluation are significant. But this effect is also influenced by the spatial resolution of the model grid: a higher resolution of the grid would lead to more variability in fitness values and hence to a more precise initialization. At the same time, a higher resolution would increase the computational demands of the initialization method. We strongly recommend to use either the geometry or the area based fitness function for the evaluation. In this study, we only take the observation of the year $t_e$ into account. Multi-temporal outlines are likely to greatly reduce uncertainties at prior times.

Our results are relevant for future glacier evolution modeling studies, as they indicate that at least for some glaciers the time needed to converge to a similar evolution regardless of the 1850 state is comparatively short. Our study might also be useful to determine a good starting point of a past simulation, e.g. to improve the initialization date in Zekollari et al. (2019). A correlation analysis of the reconstructability and glacier characteristics showed the position of the ELA, as well as the slope (especially in the lower part of the glacier) influence the reconstructabilty, whereas attributes like the glacier size do not have a strong impact. We could also show that the reconstructabilty measure correlates well with an separately obtained response time of the glacier.

Future work will include the application of the method on real-world cases, which will come with additional challenges. For example, we will have to consider the merging of neighboring glaciers when growing. Importantly, the effect of uncertainties

in the boundary conditions (in particular the glacier bed, its outlines and uncertainties in the climate forcing) will have to be quantified. This also includes to test the influence of the choice of climate conditions on the accuracy of our method. Here again, the synthetic framework will be useful by allowing data denial and data alteration experiments. To ensure the robust reconstruction of real-world glacier states additional changes and model developments are necessary. This includes e.g. the development of a glacier-individual calibration method for dynamical parameters (e.g. sliding parameter, creep parameter) as well as of the mass-balance model.

*Code availability.* The OGGM software together with initialization method are coded in the Python language and licensed under the GPLV3 free software license. The latest version of the OGGM code is available on Github (https://github.com/OGGM/oggm), the documentation is hosted on ReadTheDocs (http://oggm.readthedocs.io), and the project webpage for communication and dissemination can be found at http://oggm.org. The OGGM version used in this study is v1.1. The code for the initialization module is available on Github (https://github.com/OGGM/initialization).

## Appendix A: Temperature bias for the synthetic experiments

For the generation of the synthetic experiment state in 1850, we use a temperature bias of -1 K in order to create a relatively big glacier state. To justify the choice of this value, we have tested different temperature biases: the results are summarized in Fig. A1. This figure shows that applying positive or small negative temperature biases to the synthetic experiments results in large area differences to the RGI in 2000, and the total glacierized area in 1850 is also too small. The sample size is reduced, because less glaciers fulfill the area threshold criteria of 0.01 km$^2$. Negative temperature biases that are too large also reduce the sample size, because some runs fail (the glacier gets larger than the underlying grid). The experiments with a temperature bias of -1K, -1.25K or -1.5K perform best regarding the area difference to the RGI in 2000. But only the experiment with the temperature bias of -1K performs good regarding the estimation in 1850 of Zemp et al. (2006), whereby it needs to be taken into account that the dots only represent a subset (the small glaciers in 2000 are missing) of the glaciers considered in (Zemp et al., 2006).

## Appendix B: Initialization at different starting times

We applied our method to different starting times (1850, 1855, ..., 1965) to test how far one can go back in time to get a good initial state for this glacier. While this inverted setup is computationally very expensive, unfortunately it does not lead to improved results. See Fig. B1 for two different examples. For each tested starting year, we determined the median state and conducted an uncertainty analysis (similar to the one in Sect. 4.1). We find that the uncertainties of the median states at the different starting points are higher than doing the initialization for the year 1850 (only) and running this state forward in time. While this is counter-intuitive, the main reason is that by starting in 1850 even with a very large number and range of candidates, the very unrealistic ones are quickly forced to converge by the boundary conditions (i.e., by climate), effectively

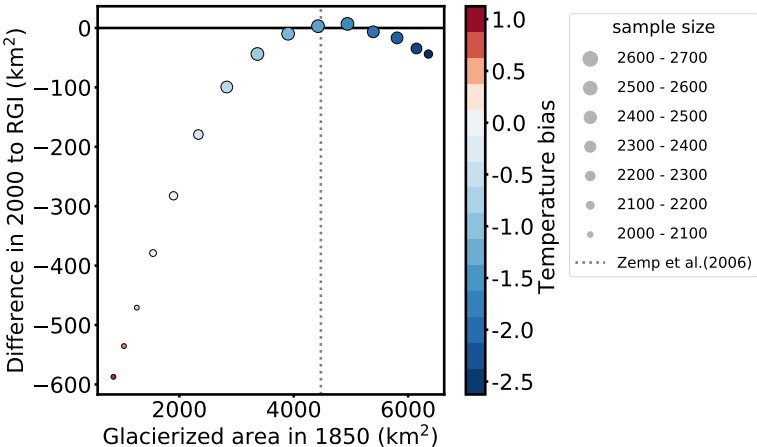

**Figure A1.** Difference between the total area in 2000 to the total area from the RGI plotted as a function of total area in 1850. Colors mark the applied temperature bias to create the synthetic experiments, and the size of the points mark the sample size (number of glaciers with an area larger than 0,01 km$^2$ in 2000). The dashed grey line marks the estimated total area of all Alpine glaciers in 1850 from (Zemp et al., 2006)

reducing the number of potential candidates for a later date. In other words, we make use of our knowledge about past climate to reduce the number of candidates at each later stage. In real-world applications, results might be different since uncertainties in past climate are large. While this should be explored further, because of the computational cost it is hard to imagine an eventual applicability on the global or even large regional scale.

5   *Author contributions.*   JE is the main developer initialization module and wrote most of the paper. BM and FM are the initiators of the OGGM project and helped to conceive this study. FM is the main OGGM developer and participated in the development of the initialization module.

*Competing interests.*   The authors declare that they have no conflict of interest.

*Acknowledgements.*   BM, and JE were supported by the German Research Foundation, grants MA 6966/1-1 and MA 6966/1-2. We thank the editor Andreas Vieli and the two anonymous referees for their comments that helped to improve our manuscript a lot.

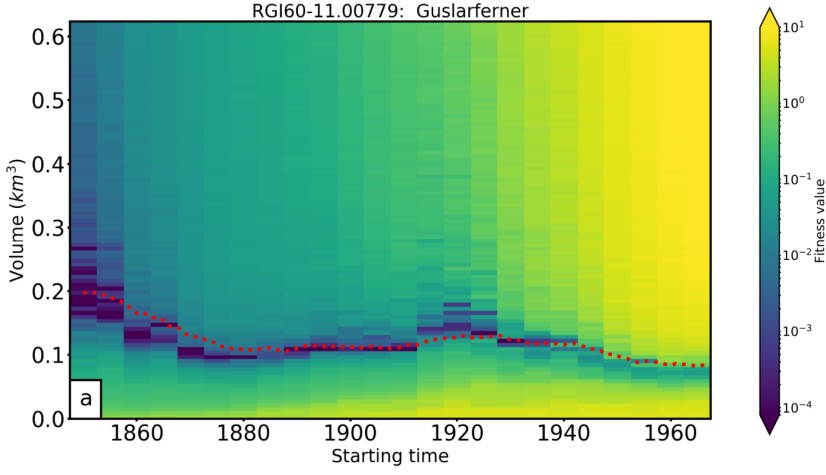

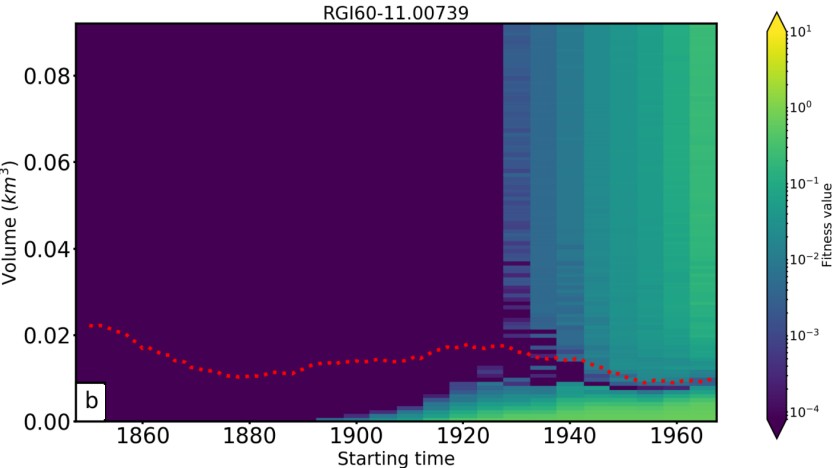

**Figure B1.** Reconstructability for different starting times. Colors indicate the fitness value of a simulation initialized with a glacier volume indicated by the vertical axis at a time indicated by the horizontal axis. Red dotted line shows the synthetic experiment. Upper panel: example for a glacier with ordinary reconstructive power; the "observed" glacier state in 2000 constrains the past evolution well in the 20th century, and the reconstruction is close to the goal. Lower panel: example for a glacier with very low reconstructive power; the "observed" glacier state does not constrain the past glacier evolution before approx. 1930.

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
