# Peer review of "Initialization of a global glacier model based on present-day glacier geometry and past climate information: an ensemble approach"

_The Cryosphere, 2019_

## Referee Comment (RC1) · Anonymous Referee #1 · 3 Jul 2019

**Review of "*Initialization of a global glacier model based on present-day glacier geometry and past climate information*"**

by J. Eis, F. Maussion and B. Marzeion

Manuscript for review in *The Cryosphere Discussions*

July 2019

In this manuscript, Eis et al. use a numerical approach to assess how well past glacier geometries can be reconstructed by relying on the present-day glacier geometry. For this purpose, they utilize the Open Global Glacier Model (OGGM), which is a state-of-the art glacier evolution model that has the capacity to model a large ensemble of glaciers (Maussion et al., 2019). The OGGM is used to simulate the transient evolution of >2000 glaciers in the European Alps between 1850 and present-day, after which they are compared to observed geometries (or rather synthetic geometries close to these). In many cases, different initial geometries lead to an almost identical present-day state (i.e. various initial states lead to unique present-day geometry). The authors also show that when using the entire information about the present-day geometry, the uncertainty on past conditions reduces compared to more simplified approaches in which only the present-day length is considered.

I must say that I was very enthusiastic when starting to read this manuscript, but that in the end I have several questions - some more substantial than others. On the hand, I think the idea is very interesting, the model is the correct one to tackle this particular problem, and the presentation of the results is neat: the text is generally easy to follow, and so are the figures. On the other hand, I have some reservations concerning the experimental setup (with a largely theoretical focus, but no incorporation of real data/observations) and the conclusions drawn from this. I have detailed on this in the next section, and hope that the authors will be able to address (some of) the issues raised. There may be some elements/passages that I may have misunderstood, and on which I would gladly be corrected, but in that case I am afraid they may also be problematic to understand for some other readers.

**General comments**

- When going through this manuscript, the first thing that popped up in my mind is: 'these experiments are all about glacier response time'. Also when reading the entire manuscript, this idea persisted: this is a response time story! I was therefore rather surprised to not see any discussion on this, or not even having it mentioned anywhere. In the end - to me - it boils down to: you can say something about the past glacier geometry (when considering the present-day geometry) over time periods that are close to or shorter than the glacier response time (depending on which definition is used for the response time). Or formulated differently: the present-day geometry does not depend on the past glacier geometry when considering time periods that exceed the glacier response time. As the response time of Alpine glaciers is typically in the order of years to decades (e.g. Haeberli and Hoelzle, 1995; Oerlemans, 2007; Zekollari and Huybrechts, 2015) it is difficult to picture how the present-day geometry (or a simulated geometry resembling this) can be used to say something about the glacier geometry in 1850.

- Continuing on the above point, do you think that what you derive as the past geometry in 1850 is realistic and that for cases where a non-unique answer (i.e. a non-unique glacier geometry) arises for the present-day: that the 'best' 1850 geometry that you obtain is really an indication of the past geometry? Are there not other model uncertainties that play a bigger role?

- You mention that you cannot perform tests with real present-day geometries. When reading the manuscript, it does not entirely become clear to me why that is. Could you elaborate on this? It would have made it

really interesting if you could have worked with real present-day geometries and performed your simulations based on this, which I was in fact what I was expecting... e.g. (1) reproduce geometries at the end of the LIA and compare these to real geometries at that time or (2) for instance compare the length changes modelled between 1850 and the present-day with observed length changes over this time period (e.g. from Leclercq et al., 2014). Such a validation would really have been of great value here, and would probably be the best way to increase our confidence in the applicability of the method you propose.

- The ideas put forward in this manuscript are really interesting, but it would be good to have a different light shining on this. I leave it up to the editor whether it is necessary to rethink/redo some of the experiments or not, but at this point, I am left with many questions and cannot help myself thinking there are some missed opportunities. Based on the above discussion, some points that could be included are for instance:

  1. Why not invert the setup? Could potentially be very interesting: for every individual glacier the question could be asked 'how far can you can back in time to have an initial geometry that still determines the present-day glacier geometry?': this is a kind of response time for every glacier. This information could be used to infer something about the response time of Alpine glaciers, even in the rather theoretical framework that you are using so far (with little to no observational data).. Again, I understand that this would be a considerable amount of work at this stage, so I am not saying that this has to be included, but I think it would be a nice addition, which would be more valuable for the reader (vs. the so-far used rather technical setup..)

  2. In whichever way the manuscript/experiments are reorganised: a discussion on the role of the response time seems crucial!

  3. Would really be rewarding to have the whole story a bit less theoretical and more applied to the real world. You model the glaciers in the European Alps, for which there is an amazing dataset on past changes...

**Specific and technical comments**

**Abstract**

- p.1, l.7: 'alpine'. Is the term 'alpine' not referring to the type of glacier (i.e. a mountain glacier) instead of the region ('Alpine')? See e.g. nsidc.org/cryosphere/glossary/term/alpine-glacier . From my understanding, here, and throughout the manuscript, you want to refer to 'Alpine' glaciers?

- p.1, l.13-15: the problem does indeed turn out to be 'non-unique' in many cases. As said earlier, even for the cases where a small difference exists in the modelled present-day glacier, I am not really convinced if this can really be seen as an indicator for how the glacier was in 1850... Could be valuable when considering shorter time periods between model initialization and present-day (e.g. multidecadal timescale), but not convinced over time periods >100 years... See also suggestion 1 at the end of the 'General comments' section.

**1 Introduction**

- very nice introduction overall. Summarizes the problem really well.

- p.1, l.18-19 and/or p.2, l.2-3: could consider adding some new references here: Zemp et al. (2019) and Wouters et al. (2019)

- p.2, l.12-13: from this: sounds like in most numerical experiments the starting point is an observed geometry, which is typically not the case (e.g. due to problems related to model drift,...etc) (e.g. Goelzer et al., 2018, which you mention later). Could consider reformulating this.

- p.2, l.21-23: do not know the details about this study, but I am surprised that unique glacier area in the past can be used to have correct glacier area at present over such long time periods. Is this an artefact of using the V-A scaling method? Can imagine that in reality, can build up same glacier when starting from two very different states in 1850 (e.g. from zero ice thickness in 1850 and when using the present-day geometry as the 1850 state). Could you comment on this here?

- p.2, l.30-31: starting in 1990 for all simulations is indeed somewhat arbitrary. Your setup could potentially be used to suggest a good starting point for every individual glacier, which could be a few decades in the past (suggestion 1 in 'General Comments' section). But again, not convinced you can go back to >100 years when it comes to deriving past glacier information from the present.

- p.2, l.32: 'most work focuses on estimating the present-day state of ice sheets'. Is a bit strangely formulated, as you make it sound like the main goal of ice sheet modellers is to correctly represent the present-day geometry of an ice sheet. Would rather say that their work is focused on making accurate projections of future ice sheet change, for which accurate reconstructions of the present-day state are crucial.

- p.2-3, l.34-35 - l.1: odd sentence. Consider splitting up / reformulating?

**2.1 The Open Global Glacier Model**

- p.3, l.25: resolution varying from 10 to 200 meters. Which criterion is used to decide which horizontal resolution to use for a given glacier?

- p.4, l.9: 'realistic climate settings': what do you mean with 'realistic' here?

**2.2 Problem description**

- p.5, l.1-2: 'such that the forward modelled state is as close as possible to the observation': OK. Here I get a bit lost however. If I understand it correctly from reading the text further, in the end you try to model a present-day state that is as close as possible to the observation, but for this purpose, as a reference/observed state, you do not use the observed present-day glacier, but a state resembling this ('observed state' of the synthetic experiments). Correct? If so, can you explain why that is? I seem to have missed this piece of information, and find it quite confusing. If not, I am still afraid that it will not be easy to understand for other reader. Potentially consider reworking this..

- p.5, l.11-12: 'non-unique': of course → related to the fact that the time period (1850 - present-day) considered exceeds the response time of this individual glacier. By here, would have expected to have the response time mentioned somewhere...

**2.3 Synthetic experiments**

- I get somewhat lost here (in general in section 2.3 and 2.4).. See first comment on previous section: is the real present-day geometry used as the 'observed state': why (not)?

- p.6, l.1-2: temperature bias of -1 K: seems rather arbitrary. For which other values was it tested? What is influence if other value would have been chosen?

- p.6, l.4: Guslarferner. Please state where this glacier is located. Why are these two Austrian glaciers (together with Hintereisferner) considered and not others? As no data is used for evaluation/calibration, it seems that any glacier could have chosen (i.e. also glaciers that are not monitored / glaciers that are less well known).

**2.4 Reconstruction of initial glacier states**

- Figure 2: from panel a, the temperature bias seems to be varying between -3.0 and 1.95 K. But in the legends a range from -2.65 to 2.95 K is mentioned. Should this be the same or am I misunderstanding this?

- p.7, l.11: 'where all trajectories have reached this stagnation period': could you be more specific? How do you define stagnation?

- p.7, l.11-12: linking the 'stagnation period' (somehow related to glacier response time) to glacier size → questionable. Not always the case that 'longer/larger glacier has got a longer response time' (e.g. Leysinger Vieli and Gudmundsson, 2004; Bahr et al., 1998; Pfeffer et al., 1998; Raper and Braithwaite, 2009; Oerlemans, 2012). It is OK to take the largest glacier to determine the stagnation period, but would be careful with the statement linking the response time to the glacier size/thickness (Jóhannesson et al., 1989)

- p.8, l.1: smoothing. Over which time period?

- p.8, l.3-4: 'equilibrium with the climate around 1850': questionable, as many Alpine glacier were still advancing and reached a maximum extent later, while others were already retreating by then (Leclercq et al., 2014). Not a major issue, but would be good if could shortly discuss this assumption.

- p.8, l.10: 'same model parameters set': could you provide a bit more details here. For instance, how is the ice rheology described (deformation/rate factor) for every glacier. How is this determined/tuned? Is this the same for every glacier? And what is the effect of this on your results?

- p.8, l.15: 'sorted by their fitness value': what do you mean by this?

**3 Test site and Input Data**

- p8, l.20: 90 m resolution DEM: this seems to be relatively low, especially given the fact that you consider glaciers as small as 0.01 km$^2$ (1 grid cell at this resolution).

- p.8, l.22: RGI date: 2003. Is the case for most glaciers in RGI6.0 (those derived by Paul et al., 2011), but not for all.

- p.8, l.24: '5 minutes resolution'. What is this approx. in meters here?

- p.8, l.27: 'threshold for the RGI': for this region? Not sure, but thought this was region-dependent?

**4 Results**

- p.9, l.7: non-uniqueness for Guslarferner: as expected, given the fact that response time of this glacier is likely much shorter than the period between 1850 and present day... Do not think you can really say anything about glacier state in 1850 from these experiments, as you also point out. But also for Hintereisferner (p.9, l.25-...): not convinced that the narrower set of 1850 sets having a lower 'fitness value' are indicative for how the glacier was in 1850..Intuitively, expect that you can say something about past Hintereisferner state (from present-day state) for max. a few decades back in time (1950 maybe?)

- To be more convincing, would really be rewarding to compare these 'best' estimates for the 1850 state with observations. And this ideally for a large set of glaciers. Think this could work, also for the length changes between 1850 and present-day, even when considering the fact both states (1850 and present-day) are synthetic → are the length changes close to observations for your 'best' results (i.e. closer when considering the 1850 states that are not being considered as 'good')?

- p.9, l.10: '...200 candidates has a...' → '...have a...'

- p.9, l.16-17: 'in close proximity to the synthetic experiments': OK. Because do not compare to the real observation. Is it not possible to work with real observation? To get match here, would probably have to tune some model parameters. Is this the problematic aspect of working with observed present-day geometry? (seems to be the case when reading the 'Discussion and conclusions' section)

- p.9, l.18: 'the observation': which is also modelled, right?

- p.9, l.25: results are different: quite trivial. Would reformulate this or simply omit this

- p.9, l.26: only few with small fitness value: see first comment on this section

- p.9, l.27-28: 'need more time to adapt' → response time!

- p.10, l.1: '11.7 to 12.4 km': from this: deduce a retreat of ca. 5 km. Is this the case? Would be really interesting to compare. For this glacier and for many others. Would make story much stronger.

- p.10, l.6: 'we were able to show': personally not really convinced at this point unfortunately. See suggestion about incorporating 'real' data (observed past states and observed (length) changes)

- p.10, l.7-8: combining with climatic information. How big is the role of the past climatic information on your results? Does the choice of past conditions (the conditions that you impose) affect the 'best' past states?

- second part of section 4.1 and section 4.2: well presented!

- Figure 9+10: number of glaciers mentioned in figure title = 2619 vs. 2621 before. How come?

- p.15, l.1: 'This holds also true': strange formulation. Consider reformulating

**5 Hardware requirements and performance**

- Very nice to have such a section. Very useful to get an insight into this!

- p.15, l.12: 'influence strongly' → 'strongly influences'

**6 Discussion and conclusions**

- p.16, l.3-4: 'identify errors...introduced by model approximation': OK, but this is not very satisfying for the reader and makes the paper almost purely theoretical...

- p.8-10: synthetic glacier state in 2000. Due to this cannot say something about real glacier state in 1850. Is it not possible to say something about the (length) change during this period and compare this to observations (for the 'best solutions')?

- p.13: 'which we don't address here' → 'which we do not address here'. A pity...I understand that this is difficult, as explained by the authors (p.16, l.4-7), but question may arise at this point how relevant this story is for the community? Seems to be a good starting point for further research by the group and users of OGGM (p.17, l.26-27: 'framework will be useful'), but more limited for outsiders. Feels more like a 'technical' note/paper now. Here again, I am convinced that by incorporating some 'real data' and/or rethinking of some experiments (see e.g. also suggestion 1 in 'general comments' section): the story would become far more relevant for the glacier (modelling) community.

- p.16, l.15: 'observed state': really the observed state or a synthetic one?

- p.16, l.16: 'depends on the situation' → 'depends on the specific glacier setting'?

- p.17, l.8: 'performs equally good' → 'performs equally well'?

- p.17, l.9: 'In Sect 4.2...': find it strange to have this formulated in such a manner in the conclusion

- p.17, l.12: 'could differ strongly in volume and area': would expect the differences to be relatively small with a trapezium/parabola cross-section, no?

- p.17, l.13: 'lead more variability' → 'lead to more variability'

- p.17, l.18: 'for some glaciers': indeed, for the ones with a short response time

**References**

Bahr, D. B., Pfeffer, W. T., Sassolas, C., and Meier, M. F.: Response times of glaciers as a function of size and mass balance: 1. Theory, Journal of Geophysical Research, 103, 9777–9782, doi:doi:10.1029/98JB00507, 1998.

Goelzer, H., Nowicki, S. M., Edwards, T. L., Beckley, M., Abe-Ouchi, A., Aschwanden, A., et al.: Design and results of the ice sheet model initialisation experiments initMIP-Greenland: an ISMIP6 intercomparison, The Cryosphere, 12, 1433–1460, doi:10.5194/tc-12-1433-2018, 2018.

Haeberli, W. and Hoelzle, M.: Application of inventory data for estimating characteristics of and regional climate-change effects on mountain glaciers: a pilot study with the European Alps, Annals of Glaciology, 21, 206–212, doi:10.3189/S0260305500015834, 1995.

Jóhannesson, T., Raymond, C. F., and Waddington, E. D.: A simple method for determining the response time of glaciers, in: Glacier Fluctuations and Climatic Change, pp. 343–352, Kluwer Academic Publishers, 1989.

Leclercq, P. W., Oerlemans, J., Basagic, H. J., Bushueva, I., Cook, A. J., and Le Bris, R.: A data set of worldwide glacier length fluctuations, The Cryosphere, 8, 659–672, doi:10.5194/tc-8-659-2014, 2014.

Leysinger Vieli, G. J.-M. C. and Gudmundsson, G. H.: On estimating length fluctuations of glaciers caused by changes in climatic forcing, Journal of Geophysical Research, 109, F01 007, doi:10.1029/2003JF000027, 2004.

Maussion, F., Butenko, A., Champollion, N., Dusch, M., Eis, J., Fourteau, K., et al.: The Open Global Glacier Model ( OGGM ) v1.1, Geoscientific Model Development, 12, 909–931, doi:10.5194/gmd-12-909-2019, 2019.

Oerlemans, J.: Estimating response times of Vadret da Morteratsch, Vadret da Palü, Briksdalsbreen and Nigardsbreen from their length records, Journal of Glaciology, 53, 357–362, doi:10.3189/002214307783258387, 2007.

Oerlemans, J.: Linear modelling of glacier length fluctuations, Geografiska Annaler, Series A: Physical Geography, 94, 183–194, doi:10.1111/j.1468-0459.2012.00469.x, 2012.

Paul, F., Frey, H., and Bris, R. L. E.: A new glacier inventory for the European Alps from Landsat TM scenes of 2003: challenges and results, Annals of Glaciology, 52, 144–152, doi:10.3189/172756411799096295, 2011.

Pfeffer, W. T., Sassolas, C., Bahr, D. B., and Meier, M. F.: Response time of glaciers as a function of size and mass balance: 2. Numerical experiments, Journal of Geophysical Research, 103, 9783–9789, 1998.

Raper, S. and Braithwaite, R.: Glacier volume response time and its links to climate and topography based on a conceptual model of glacier hypsometry, The Cryosphere, 3, 183–194, doi:10.5194/tcd-3-243-2009, 2009.

Wouters, B., Gardner, A. S., and Moholdt, G.: Global Glacier Mass Loss During the GRACE Satellite Mission (2002-2016), Frontiers in Earth Science, 7, 96, doi:10.3389/feart.2019.00096, 2019.

Zekollari, H. and Huybrechts, P.: On the climate-geometry imbalance, response time and volume-area scaling of an alpine glacier: insights from a 3-D flow model applied to Vadret da Morteratsch, Switzerland, Annals of Glaciology, 56, 51–62, doi:10.3189/2015AoG70A921, 2015.

Zemp, M., Huss, M., Thibert, E., Eckert, N., McNabb, R., Huber, J., et al.: Global glacier mass balances and their contributions to sea-level rise from 1961 to 2016, Nature, 568, 368–386, doi:10.1038/s41586-019-1071-0, 2019.

---

## Referee Comment (RC2) · Anonymous Referee #2 · 11 Jul 2019

Review of the manuscript entitled:

**Initialization of a global glacier model based on present-day glacier geometry and past climate information: an ensemble approach**

Julia Eis et al.

**Summary**

The authors present an initialisation strategy for the Open Global Glacier Model (OGGM). The aim of this strategy is twofold. First, the initialisation should produce a best estimate for the glacier extent and geometry at the end of the little ice age (LIA). Second, the spin-up into present day should appropriately reproduce the observed geometry. The latter aim is formulated as an optimisation of an inverse problem. For testing and validation of the initialisation, the authors suggest synthetic experiments for which the past and present geometry is perfectly known. The main target parameter for the optimisation is a so-called 'temperature bias' $\beta$. The authors show that their strategy allows to constrain the possible parameter space significantly, certainly if the optimisation accounts for geometric information beyond the glacier length.

The manuscript is of great interest to the community as it aims at formulating a standard procedure that should provide the initialisation basis for regional or global ice-dynamic simulations of glacier evolution. In this sense, the authors are trying to solve a pressing issue. Moreover, the manuscript is well written and structured. However, I have major concerns on the pursued parameter sampling strategy in light of other a-priori parameter choices concerning ice-dynamics and surface mass balance. Consequently, I fear that the single parameter problem is oversimplifying any real-world application. To address my comments, the authors will certainly have to expand the manuscript to better justify and motivate their decisions. Consequently, I recommend a major revision of the manuscript.

**General comments**

**Temperature bias**
For the synthetic experiment, you prescribe a 'random climate scenario'

(p5l26). I think that this term refers to a random permutation
of the climatic forcing around the year 1850 (within a 30 year period).
Is that right? I assume that the climatic forcing is taken from
HISTALP. You then add a temperature bias of -1K to this climatic
forcing. This bias is not well motivated. Why is it necessary?
Initially, I thought that HISTALP provide you with the perfect climatic
conditions. On second thought however, I see many reasons why this
bias is necessary. This point is very important as the temperature
bias serves as on of the main target quantities for the subsequent
optimisation/data assimilation. Please provide a firm motivation
for this bias, why it varies from glacier to glacier and consequently
has to be inferred.

**A-priori OGGM calibration vs. parameter sampling**
Initially, the first question which came into my mind was why you
did not enlarge the ensemble by including other uncertain parameters,
as for instance the rate factor, basal friction or parameters linked
to the SMB model. So ultimately this question links to the comment
above. In other words, how do you ensure that the other parameters
are well constrained. I understand that you present a synthetic
setup but with regard to any real-world application this issue is
very important. Even if there was an a-priori calibration of a
combination of the rate factor and the melting parameters, other
combinations might also produce plausible glacier geometries at
present. Yet the exact choice will affect the past glacier geometry.
In many other glaciological applications, basal friction or the
rate factor are the central unknowns that are calibrated during
the model initialisation. I therefore think that it is inevitable
that you include a section on the OGGM procedure for the calibration/choice
of these other parameters. On this basis, you should motivate your
parameter choice for you ensemble generation. Dependent on how
the other OGGM parameters are calibrated, it might well be necessary
to include further parameters in the ensemble generation. At the
moment, two climatic quantities are varied: the permutation of
the climatic forcing and the temperature bias. Throughout the manuscript,
I sensed some redundancy because I did not find any discussion of
the 'acceptable ensemble states' in terms of the initial parameter
choice. To put it bluntly, were you able to infer the -1K temperature
bias prescribed in the synthetic experiment from your data assimilation?
Otherwise, the specific climatic permutation might have had a significant
influence? In short, please justify your choice to exclusively
focus on climatic quantities in the optimisation. From my experience,
you should include other parameters in this optimisation. If you
disagree, please provide good arguments for you choice.

I wonder about the necessity to permute the initial climate time series during the initialisation. Do you really attain distinctly different glacier geometries in 1850 that you could not generate by only changing $\beta$. Please re-assess the dual sampling of climatic variables/input in the ensemble generation.

**Climatic forcing**
For the generation of an ensemble of initial states, two quantities are varied. On the one hand, the climatic forcing around 1850 is permuted temporally within a 30 year period. On the other hand, an offset temperature bias $\beta$ is varied within -2K to +2K. Once the 'equally space' ensemble members are selected, forward simulations are start using climatic forcing from the HISTALP record. The past-to-present volume evolution of Guslaferner of all ensemble members is shown in Fig. 3. It suprised me that all ensemble members readily converge to a very similar present-day value. Certainly if you consider the large range of $\beta$-values used for the initialisation. On page 8 lines 10- 11, my attention was then drawn to the fact that all forward runs are conducted with 'exactly the same climate time series' and use the 'same parameter set'. Does this include $\beta$? If not, why does the -2K ensemble memebr not stick out in terms of present ice volume. If $\beta$ was set to zero after the initialisation, I would be highly concerned about the abrupt climatic shift you introduce when switching from the initialisation to the forward experiment. A reason for the latter case is the overall quick convergence during the forward simulations. Supporting evidence comes from a formulation on page 6 line 2 where you say that $\beta$ is an initial bias, invoking that it is set to zero in the forward simulations.

In any case, this question directly relates to my second concern on the a-priori calibration of OGGM in terms of the SMB module. What climatic forcing is used for the a-priori OGGM calibration. Is it HISTALP. Is the temperature bias $\beta$ included? Please clarify.

**Detailed comments**

**P1L20** Sentence is difficult to understand. Reformulate.
**P2L17** 'Despite of the importance ...' --> 'Despite the importance ...'
**P2L28** Introduce a comma (,) before which. Please check throughout the document.

**P3L1** '... glacier's length ...' --> '... glacier length ...'.
Please avoid the 's genitive throughout the document.
**P5L5** Please omit the bedrock difference in Equation (4). As the
bedrock does not change during your initialisation this term is
zero.
**P5L22** Remove comma.
**P5L28** Delete 'this' at the beginning of the line.
**P6L11-P7L5** The initialisation ensemble is formed by the variation
of two parameter: $\beta$ and a permutation of the 30-yr climate forcing
time series. For both quantities it remains unclear how many ensemble
members are created. As you reduced the ensemble later on to 200
members based on an equal spacing argument, I would assume a sufficiently
dense sampling. Anyhow, please provide some numbers.
**P6L13** It remains vague how the permutation is done. You permute
the climatic forcing per year, month, etc.
**P9L6** Please provide values for glacier area and estimates for mean
ice thickness for Guslarferner and Hintereisferner at present. Hinterisferner
covers a much larger area and is probably much thicker. These values
are informative and they are difficult to infer from Figs. 3 and
5. **P9L10** 'has' --> 'have'
**P10L2** 'not' --> 'no'

**Figure 2** There is a discrepancy between the range of temperature
biases given in the caption [-2.65, 2.95] and shown in panel (a)
[-3.0, 1.95]. Moreover, I would try to remove some redundant information.
For panel (b), limit the graph to the stagnation period. For panel
(c), do show the initialisation period with the coloured points
but extend the figure by the forward simulation 1850-2000 as in
Fig3c.

---

## Author Comment (AC1) · 27 Aug 2019

**Point by point response to the reviewers**

We would like to thank you for your constructive feedback on our manuscript. We have done our best to address the points raised by the reviewers and hereby submit a new version of the manuscript together with a point by point reply to each of reviewers' comments. After answering questions that were mentioned from both reviewers, we provide a point by point answer to the reviewers' comments. ("RC" stands for "reviewer comment", "AR" for "authors response")

**Synthetic experiments**

After going trough the reviews, we realized that most points raised by both reviewers lead back to the design of the synthetic experiments (on which our paper is based) and a better explanation of their benefits is required to make the the relevance of our study more clear.

Reconstructing past glacier states is a complex inverse problem, and the results will depend on (i) the uncertainties involved in the boundary conditions (climate, glacier bed, etc.), (ii) the uncertainties in the glacier model itself (as pointed by reviewer #2), and (iii) a theoretical lower bound (termed "reconstructability" in our manuscript and associated to the concept of "response time" by reviewer #1) tied to the characteristics of the glacier itself (slope, size, the past climate, etc.) as well as how much we know about today's glacier and how far in time we attempt to do our reconstruction. In our opinion, addressing all three issues at once is too much for a single study.

The main point of the synthetic experiments is to separate these issues from one another, and to focus on point (iii) only. Thanks to the synthetic experiments, we are able to isolate and understand the limitations and errors of our method itself, as opposed to uncertainties that derive from unknown boundary conditions and model parameters. They also allowed us to detect what kind of observations are necessary to reduce the uncertainties of our reconstruction (for example, that length observations are less suitable than area, or better: topography).

That being said, we agree with the points raised by both reviewers and the editor, and modified our manuscript by explaining our objectives more clearly and adding a new results section and more discussions. Importantly, we now better show that although theoretical, our study also has real-world applications by providing insights about the glacier themselves. We learn from our study what makes a glacier "reconstructable", and discuss how this knowledge can be transferred to the real world. Our synthetic experiments are designed to have a certain similarity with the real world, so that we can expect to learn something meaningful from them: a point which we didn't explain clearly in our first manuscript and we hope to have improved now.

**Minor changes to the method**

During the review process, we realized that our mass balance model did not perform as good as it could have. We used OGGM default values for mass balance specific parameters, which were calibrated with a global climate dataset (CRU). We now use a

new parameter set which was determined by a cross-validation for the Alps only (41 reference glaciers) using HISTALP data. We updated this parameter set to: precipitation scaling factor $p_f = 1.75$ (instead of 2.5), melt temperature $T_{Melt} = -1.75°C$ (instead of $-1$), liquid precipitation temperature $T_{Liquid} = 2.0°C$ (unchanged) and temperature lapse rate $\Gamma = -6.5$ K km$^{-1}$ (unchanged). The new mass balance calibration lead to some changes, as the model is sensitive to this choice. First, the glaciers'

5   response times increased a little bit. We therefore had to extend the 400 years random climate runs to 600 years, to ensure that for every case $t_{stag}$ can be properly determined. Also, the number of experiments that fulfill the area threshold of $0.01$km$^2$ is enlarged to 2660 glaciers thanks to this change. We updated all figures and numbers in the manuscript without any significant change to our conclusions.

**1   Reply to Reviewer #1**

10   "RC: *In this manuscript, Eis et al. use a numerical approach to assess how well past glacier geometries can be reconstructed by relying on the present-day glacier geometry. For this purpose, they utilize the Open Global Glacier Model (OGGM), which is a state-of-the art glacier evolution model that has the capacity to model a large ensemble of glaciers (Maussion et al., 2019). The OGGM is used to simulate the transient evolution of > 2000 glaciers in the European Alps between 1850 and present-day, after which they are compared to observed geometries (or rather synthetic geometries close to these). In many cases,*

15   *different initial geometries lead to an almost identical present-day state (i.e. various initial states lead to unique present-day geometry). The authors also show that when using the entire information about the present-day geometry, the uncertainty on past conditions reduces compared to more simplified approaches in which only the present-day length is considered. I must say that I was very enthusiastic when starting to read this manuscript, but that in the end I have several questions - some more substantial than others. On the hand, I think the idea is very interesting, the model is the correct one to tackle this particular*

20   *problem, and the presentation of the results is neat: the text is generally easy to follow, and so are the figures. On the other hand, I have some reservations concerning the experimental setup (with a largely theoretical focus, but no incorporation of real data/observations) and the conclusions drawn from this. I have detailed on this in the next section, and hope that the authors will be able to address (some of) the issues raised. There may be some elements/passages that I may have misunderstood, and on which I would gladly be corrected, but in that case I am afraid they may also be problematic to understand for some other*

25   *readers.*"

**AR:** We would like to thank you for your detailed feedback and the many suggestions to improve our study. We hope that the following answers and changes in the manuscript sufficiently address your reservations concerning the experimental setup.

**General comments**

"RC: *When going through this manuscript, the first thing that popped up in my mind is: 'these experiments are all about glacier*

30   *response time'. Also when reading the entire manuscript, this idea persisted: this is a response time story! I was therefore rather surprised to not see any discussion on this, or not even having it mentioned anywhere. In the end - to me - it boils down to: you can say something about the past glacier geometry (when considering the present-day geometry) over time periods that are*

*close to or shorter than the glacier response time (depending on which definition is used for the response time). Or formulated differently: the present-day geometry does not depend on the past glacier geometry when considering time periods that exceed the glacier response time. As the response time of Alpine glaciers is typically in the order of years to decades (e.g. Haeberli and Hoelzle, 1995; Oerlemans, 2007; Zekollari and Huybrechts,2015) it is difficult to picture how the present-day geometry*

5  *(or a simulated geometry resembling this) can be used to say something about the glacier geometry in 1850.* "

**AR:** Indeed, we discussed the response time topic during the development of our method and manuscript. And yes, we agree that our study indirectly deals with response times. The main issue we see with this concept is that it is not well-defined and often causes misunderstandings. The most popular quantifiable value is the response time as defined by the e-folding time response to a step function (e.g. Oerlemans, 1997). This value depends on both the glacier state and the step change in climate,

10  and will change over time.

In order to quantitatively address your hypothesis (*"the present-day geometry does not depend on the past glacier geometry when considering time periods that exceed the glacier response time."*), we defined a new non-dimensional measure (termed "reconstructability") which assesses how easy or hard it is to find the correct past geometry. We now quantify and discuss which parameters influence this reconstructability, and among other things discuss the response time as requested. However, note that

15  the present-day geometry is not the only information entering the reconstruction, we also provide the model with information on the past climate evolution, which helps to constrain the reconstructions.

"RC: *Continuing on the above point, do you think that what you derive as the past geometry in 1850 is realistic and that for cases where a non-unique answer (i.e. a non-unique glacier geometry) arises for the present-day: that the 'best' 1850*

20  *geometry that you obtain is really an indication of the past geometry?* "

**AR:** We believe that there may be a slight misunderstanding here. If by "realistic" you mean "in the real world", we would like to refer to our explanation of how we designed the synthetic experiments below. If you mean, "realistic with respect to the true 1850 state as defined in our synthetic experiments": yes we do! Obviously, we can trust the 1850 glacier geometry of a unique case more than the 'best' candidate in a non-unique case. But with the help of the error analysis in Sect. 4.1, we can

25  prove that our 'best' candidate performs well despite of being slightly different than the true state. Especially when we run this state some years forward, the error converges to a negligible range. So in a case with non-unique solutions in 1850, we always should keep in mind that all acceptable states result in the present-day geometry and it might be better to "spin-up" the 'best' candidate some years for a better confidence.

30  "RC: *Are there not other model uncertainties that play a bigger role?* "

**AR:** For real world applications, definitely, but not in our experimental set-up. This is precisely what motivated our choice for a synthetic environment in the first place: since the system is known perfectly, and our possible glacier evolution scenarios depend only on their unknown initial state (and not on model or boundary condition uncertainties), we can address the research questions formulated above and in the introduction of the manuscript.

35  For real-world applications, model uncertainties will have to be accounted for, and will have to be compared to (and added to)

the theoretical lower bound defined in this study. We addressed this question by adding this point to the synthetic experiment section and to the conclusion.

"RC: *You mention that you cannot perform tests with real present-day geometries. When reading the manuscript, it does not entirely become clear to me why that is. Could you elaborate on this?* "

**AR:** Present-day real-world geometries are the result of a series of processes and past-climate that are largely uncertain. In our paper, we remove these uncertainties and ask the question: even if we know *everything* about the system , how constrained are the past geometries?

This said, it is of course our intention to make the reconstruction method applicable to the real world. We tried to clarify our intentions and remove the statement that the method cannot be applied to real-world glaciers in the revised manuscript.

"RC: *It would have made it really interesting if you could have worked with real present-day geometries and performed your simulations based on this, which I was in fact what I was expecting... e.g. (1) reproduce geometries at the end of the LIA and compare these to real geometries at that time or (2) for instance compare the length changes modelled between 1850 and the present-day with observed length changes over this time period (e.g. from Leclercq et al., 2014). Such a validation would really have been of great value here, and would probably be the best way to increase our confidence in the applicability of the method you propose.* "

**AR:** We apologize for misleading your expectations, and hope to have clarified the main purpose of our study with this revision. A comparison with real-world data does not make sense in the context of the synthetic experiments, and we believe that it should be part of a dedicated study that addresses the reconstruction uncertainties that are caused by uncertainties in the boundary conditions and by the model/parameter uncertainties. At the time of writing, we are confident that our method can eventually be applied to any land-terminating glacier, and we do evaluate the method against observational data as you suggest. However, as described above, this comes with a whole new set of problems, which need to be addressed in a separate manuscript.

"RC: *1. Why not invert the setup? Could potentially be very interesting: for every individual glacier the question could be asked 'how far can you can back in time to have an initial geometry that still determines the present-day glacier geometry?': this is a kind of response time for every glacier. This information could be used to infer something about the response time of Alpine glaciers, even in the rather theoretical framework that you are using so far (with little to no observational data).. Again, I understand that this would be a considerable amount of work at this stage, so I am not saying that this has to be included, but I think it would be a nice addition, which would be more valuable for the reader (vs. the so-far used rather technical setup..)* "

**AR:** Thank you for this idea! We tested it (see below) and found that while this inverted setup is computationally very expensive, unfortunately it doesn't lead to improved results: we applied our method to different starting times (1850,1855,...1965) and based on this, one can see how far one can go back in time to get a good initial state for this glacier. See Fig. 1 (of this response) for three different examples. For each tested starting year, we determined the median state and conducted an uncer-

[Figure]

**Figure 1.** Reconstructability for different starting times. Colors indicate the fitness value of a simulation initialized with a glacier volume indicated by the vertical axis at a time indicated by the horizontal axis. Red dotted line shows the synthetic experiment. Upper panel: example for a glacier with high reconstructive power; the "observed" glacier state in 2000 constrains the past evolution well into the 19 th century, and the reconstruction is close to the goal. Middle panel: example for a glacier with low reconstructability before approx. 1920 and high reconstructability afterwards. Lower panel: example for a glacier with very low reconstructive power; the "observed" glacier state does not constrain the past glacier evolution before approx. 1930.

tainty analysis (similar to the one in the manuscript). We find that the uncertainties of the median states at the different starting points are higher than doing the initialization for the year 1850 (only) and running this state forward in time. While this is counter-intuitive, the main reason is that by starting in 1850 even with a very large number and range of candidates, the very unrealistic ones are quickly forced to converge by the boundary conditions (i.e., by climate), effectively reducing the number of potential candidates for a later date. In other terms, we make use of our knowledge about past climate to reduce the number of candidates at each later stage. In real-world application, results might be different since uncertainties in past climate are large. We will explore this further, but because of the computational cost, it is hard to imagine an eventual applicability on the global or even large regional scale.

"RC: *In whichever way the manuscript/experiments are reorganised: a discussion on the role of the response time seems crucial!* "

**AR:** We agree. We added this discussion to the new section about reconstructability and to the conclusions as well.

"RC: *Would really be rewarding to have the whole story a bit less theoretical and more applied to the real world. You model the glaciers in the European Alps, for which there is an amazing dataset on past changes* "

**AR:** We hope that our revised formulation of the research questions better addresses how we can learn about the real world from our setup, even though our experiments are synthetic (see Sec. 4.3 in the manuscript)

**Specific and technical comments**

**Abstract**

"RC: *p.1, l.7: 'alpine'. Is the term 'alpine' not referring to the type of glacier (i.e. a mountain glacier) instead of the region ('Alpine')? See e.g. nsidc.org/cryosphere/glossary/term/alpine-glacier . From my understanding, here, and throughout the manuscript, you want to refer to 'Alpine' glaciers?* "

**AR:** We agree. We changed the term throughout the manuscript.

"RC: *p.1, l.13-15: the problem does indeed turn out to be 'non-unique' in many cases. As said earlier, even for the cases where a small difference exists in the modelled present-day glacier, I am not really convinced if this can really be seen as an indicator for how the glacier was in 1850... Could be valuable when considering shorter time periods between model initialization and present-day (e.g. multidecadal timescale), but not convinced over time periods > 100 years... See also suggestion 1 at the end of the 'General comments' section.* "

**AR:** As explained above, we don't want to imply that our reconstructions are valid for the real world. However, in the context of response time, we would argue that in particular reconstructions exceeding the response time can be expected to work better, because in this case, the climate forcing will have stronger a impact on the evolution of the glacier geometry than the initial geometry. Or, to put it differently: A glacier with a short response time will have "forgotten" the initial conditions quickly, and

the reconstructed evolution will be a result of climate over a large extent of the reconstruction; a glacier with a long response time will be influenced by the initial conditions for a longer time period, thus reducing the fraction of time where it is governed by the known climate.

**Introduction**

"RC: *p.1, l.18-19 and/or p.2, l.2-3: could consider adding some new references here: Zemp et al. (2019) and Wouters et al. (2019)* "

**AR:** We added the two references.

"RC: *p.2, l.12-13: from this: sounds like in most numerical experiments the starting point is an observed geometry, which is typically not the case (e.g. due to problems related to model drift,...etc) (e.g. Goelzer et al., 2018, which you mention later). Could consider reformulating this.* "

**AR:** We agree. We have reformulated this to make clear, that this is valid for models that derive the initial surface hypsometry from DEM's and outlines.

"RC: *p.2, l.21-23: do not know the details about this study, but I am surprised that unique glacier area in the past can be used to have correct glacier area at present over such long time periods. Is this an artefact of using the V-A scaling method? Can imagine that in reality, can build up same glacier when starting from two very different states in 1850 (e.g. from zero ice thickness in 1850 and when using the present-day geometry as the 1850 state). Could you comment on this here?* "

**AR:** On p.4, l.31 we wrote that "the backwards reconstruction is impeded by the non-linear interaction between glacier geometry, ice flow and mass balance." The non-uniqueness in our case basically stem from the SIA equation, on which OGGM is based. This equation can be transformed into a diffusion equation, for which the backwards problem is ill-posed (e.g. because of the non-uniqueness). So yes: The V-A scaling method is not based on a diffusion equation. Thus, it does not lead to non-unique reconstructed glacier areas in the past (although this has not been proven unequivocally in Marzeion et al., 2012).

"RC: *p.2, l.30-31: starting in 1990 for all simulations is indeed somewhat arbitrary. Your setup could potentially be used to suggest a good starting point for every individual glacier, which could be a few decades in the past (suggestion 1 in 'General Comments' section). But again, not convinced you can go back to > 100 years when it comes to deriving past glacier information from the present.* "

**AR:** Yes, this would be possible. We add this to the discussion.

"RC: *p.2, l.32: 'most work focuses on estimating the present-day state of ice sheets'. Is a bit strangely formulated, as you make it sound like the main goal of ice sheet modellers is to correctly represent the present-day geometry of an ice sheet. Would rather say that their work is focused on making accurate projections of future ice sheet change, for which accurate reconstructions of the present-day state are crucial.* "

**AR:** We agree, we rephrased the sentence.

"RC: *p.2-3, l.34-35 - l.1: odd sentence. Consider splitting up / reformulating?* "

**AR:** We split the sentence into two.

**5   The Open Global Glacier Model**

"RC: *p.3, l.25: resolution varying from 10 to 200 meters. Which criterion is used to decide which horizontal resolution to use for a given glacier?* "

**AR:** We here use OGGM's default: $dx = \sqrt{aS}$ with $a = 14$ and $S$ the area of the glacier in $km^2$. We added this information in the text.

"RC: *p.4, l.9: 'realistic climate settings': what do you mean with 'realistic' here?* "

**AR:** We mean that the forcing of the climate setting is realistic (because it comes from real climate data, but the order is changed). We rephrased the sentence.

**Problem description**

15   "RC: *p.5, l.1-2: 'such that the forward modelled state is as close as possible to the observation': OK. Here I get a bit lost however. If I understand it correctly from reading the text further, in the end you try to model a present-day state that is as close as possible to the observation, but for this purpose, as a reference/observed state, you do not use the observed present-day glacier, but a state resembling this ('observed state' of the synthetic experiments). Correct? If so, can you explain why that is? I seem to have missed this piece of information, and find it quite confusing. If not, I am still afraid that it will not be easy to*

20   *understand for other reader. Potentially consider reworking this.* "

**AR:** We here mean an observation in general (either the present state or the observed synthetic experiment). But we agree that this can be confusing for the reader. We have reworked the synthetic experiments section to avoid confusion.

"RC: *p.5, l.11-12: 'non-unique': of course $\rightarrow$ related to the fact that the time period (1850 - present-day) considered exceeds*

25   *the response time of this individual glacier. By here, would have expected to have the response time mentioned somewhere...* "

**AR:** The non-uniqueness stem from the ill-posedness of the backwards SIA/(diffusion problem) or in other words: because the reconstruction is impeded by the non-linear interaction between glacier geometry, ice flow and mass-balance. Please also note (as mentioned above) that reconstructions exceeding response time are potentially possible through the availability of information on past climate.

**Synthetic experiments**

"RC: *I get somewhat lost here (in general in section 2.3 and 2.4).. See first comment on previous section: is the real present-day geometry used as the 'observed state': why (not)?* "

**AR:** We clarified these terms in the manuscript. In general one could say that the method is described for an observation in general (either a real present-day geometry or the synthetic experiment). The method is also implemented in that way. The synthetic experiment in 2000 is then treated as an observation. But we will make this more clear to the reader.

"RC: *p.6, l.1-2: temperature bias of -1 K: seems rather arbitrary. For which other values was it tested? What is influence if other value would have been chosen?* "

**AR:** Setting the temperature bias to -1K is not completely arbitrary. To justify our choice, we tested different temperature biases for the generation of the synthetic experiments. The results are summarized in Fig. 2:

[Figure]

**Figure 2.** Difference between the total area in 2000 to the total area from the RGI plotted as a function of total area in 1850. Colors mark the applied temperature bias to create the synthetic experiments, and the size of the points mark the sample size (number of glaciers with an area larger than 0,01 km$^2$ in 2000). The dashed grey line marks the estimated total area of all Alpine glaciers in 1850 from (Zemp et al., 2006)

This figure shows that applying positive or small negative temperature biases to the synthetic experiments results in large area differences to the RGI in 2000, and the total glacierized area in 1850 is also too small. The sample size is reduced, because less glaciers fulfill the area threshold criteria of 0.01 km$^2$. Negative temperature biases that are too large also reduce the sample size, because some runs fail ("Glacier exceeds boundary", this means the glacier would get larger than the underlying grid). The experiments with a temperature bias of -1K, -1.25K or -1.5K perform best regarding the area difference to the RGI in 2000. But only the experiment with the temperature bias of -1K performs good regarding the estimation in 1850 of Zemp et al.

(2006), whereby it needs to be taken into account that the dots only represent a subset (the small glaciers in 2000 are missing) of the glaciers considered in (Zemp et al., 2006). We added this results to Appendix A.

"RC:*p.6, l.4: Guslarferner. Please state where this glacier is located. Why are these two Austrian glaciers (together with Hintereisferner) considered and not others? As no data is used for evaluation/calibration, it seems that any glacier could have chosen (i.e. also glaciers that are not monitored / glaciers that are less well known).* "

**AR:** We added information, where the two example glaciers are located and why we picked them as examples. And of course, different glaciers could also be chosen. For the uncertainty analysis, we run each of the 2660 glaciers and we produce the same plots for all. But for a more representative set of glaciers, we added several new examples (15) as new supplementary material.

**Reconstruction of initial glacier states**

"RC: *Figure 2: from panel a, the temperature bias seems to be varying between -3.0 and 1.95 K. But in the legends a range from -2.65 to 2.95 K is mentioned. Should this be the same or am I misunderstanding this?* "

**AR:** Yes, they should be the same. During the development of the manuscript, we changed the example glacier in this Figure and we forgot to change the numbers in the description. We corrected them.

"RC: *p.7, l.11: 'where all trajectories have reached this stagnation period': could you be more specific? How do you define stagnation?* "

**AR:** During the stagnation period the glacier volume does not increase or decrease strongly in comparison to the total volume since the beginning of the simulation. We added this information to the text.

"RC: *p.7, l.11-12: linking the 'stagnation period' (somehow related to glacier response time) to glacier size → questionable. Not always the case that 'longer/larger glacier has got a longer response time' (e.g. Leysinger Vieli and Gudmundsson, 2004; Bahr et al., 1998; Pfeffer et al., 1998; Raper and Braithwaite, 2009; Oerlemans, 2012). It is OK to take the largest glacier to determine the stagnation period, but would be careful with the statement linking the response time to the glacier size/thickness (Jóhannesson et al., 1989)* "

**AR:** We agree. We removed the link to Jóhannesson.

"RC: *p.8, l.1: smoothing. Over which time period?* "

**AR:** 10 years, we added this information to the text.

"RC: *p.8, l.3-4: 'equilibrium with the climate around 1850': questionable, as many Alpine glacier were still advancing and reached a maximum extent later, while others were already retreating by then (Leclercq et al., 2014). Not a major issue, but would be good if could shortly discuss this assumption.* "

**AR:** We agree. To a certain extent, the synthetic experiments simulate this situation since the target states are created from a

randomly fluctuating climate, and therefore some of the glaciers are advancing while others are retreating. We cannot really assess if these advances are comparable to those observed in the real world after 1850. We added a mention to advancing glaciers in the synthetic experiments description section.

5   "RC: *p.8, l.10: 'same model parameters set': could you provide a bit more details here. For instance, how is the ice rheology described (deformation/rate factor) for every glacier. How is this determined/tuned? Is this the same for every glacier? And what is the effect of this on your results?* "

**AR:** We added more information about the dynamical parameters used in this study (which are the same for every glacier) at the end of Sect. 2.1, but we refer at the same time to (Maussion et al., 2019). There you can also find a discussion about 10   the model sensitivity to these parameters (for the Hintereisferner and on a global scale). As the model is sensitive to these parameter settings, a change consequently would also lead to different ice volumes and response times in this study.

"RC: *p.8, l.15: 'sorted by their fitness value': what do you mean by this?* "

**AR:** For each glacier we return a DataFrame (similar to a table), containing information about each glacier candidate that was 15   evaluated. This DataFrame is sorted by the fitness values of the candidates. (The ones with the lowest fitness values come first). We reformulate this sentence.

**Test site and Input Data**

"RC:*p8, l.20: 90 m resolution DEM: this seems to be relatively low, especially given the fact that you consider glaciers as small as 0.01 km$^2$ (1 grid cell at this resolution).* "

20   **AR:** We agree that in these cases the resolution is low, but this is the standard procedure of OGGM and similar models, and is driven by data availability. The 90 m resolution DEM is interpolated to higher resolution grids (down to $dx = 10m$ for small glaciers), leading to a smooth field. To make sure that the glacier outline touches more than one grid cell of the DEM, OGGM test if the minimum and maximum value are equal. If they are and error will be raised and this glacier can not be modelled.

25   "RC: *p.8,l.22: RGI date:2003. Is the case for most glaciers in RGI6.0 (those derived by Paul et al.2011), but not for all*
  **AR:** We changed this.

"RC: *p.8, l.24: '5 minutes resolution'. What is this approx. in meters here?* "

**AR:** This correspond to a resolution of approx. 9,3 km. We added this information to the manuscript.

30

"RC: *p.8, l.27: 'threshold for the RGI': for this region? Not sure, but thought this was region-dependent?* "

**AR:** The threshold is the same for all regions. You can find this information in the RGI 6.0 Technical Report (e.g. page 18, Section 3.4 "Quality Control", step 2)

**Results**

"RC:*p.9, l.7: non-uniqueness for Guslarferner: as expected, given the fact that response time of this glacier is likely much shorter than the period between 1850 and present day... Do not think you can really say anything about glacier state in 1850 from these experiments, as you also point out. But also for Hintereisferner (p.9, l.25-...): not convinced that the narrower set*
5  *of 1850 sets having a lower 'fitness value' are indicative for how the glacier was in 1850.Intuitively, expect that you can say something about past Hintereisferner state (from present-day state) for max. a few decades back in time (1950 maybe?)* "

**AR:** Here, we disagree. First, we think that the concept of "response time" cannot be used to discard any climate signal longer that a few decades: Hintereisferner certainly needed more than 50 years to grow as big as it is today (i.e. today's state does contain information about past states, at minimum by discarding obvious impossible candidates). More importantly, the narrower
10  set of 1850 sets having a lower fitness value is an indication for how the glacier was in 1850, at least in our synthetic experiments - for which we know the "truth", and are able to assess the accuracy of the method. In the case of the Guslarferner, the response time is shorter than in the case of the Hintereisferner. All acceptable states need then to be considered as possible solutions, and the median state is only a good representative of this set. The uncertainties definitely need to be taken into account for these cases, but running the median state some years forward reduces the uncertainties rapidly (as shown in Fig. 7-10).
15  However, we agree that (as explained above), uncertainties in past climate conditions as well as uncertainties of the model and its parameters will increase the reconstruction uncertainty in a real-world application, such that the good reconstructabilities obtained here cannot be expected to be carried over to real-world conditions.

20      "RC: *To be more convincing, would really be rewarding to compare these 'best' estimates for the 1850 state with observations. And this ideally for a large set of glaciers. Think this could work, also for the length changes between 1850 and present-day, even when considering the fact both states (1850 and present-day) are synthetic → are the length changes close to observations for your 'best' results (i.e. closer when considering the 1850 states that are not being considered as 'good')?* "

**AR:** The synthetic experiment length changes should not be compared to observations. The 1850 glacier geometry, depends
25  strongly on the observation (here the synthetic experiment) as the candidates are evaluated by the difference to the observation. As the synthetic experiment in 2000 differ to the real-observation, a different trajectory is declared as the best candidate. This trajectory can also differ in gradient. Such a comparison only makes sense, if the candidates are evaluated based on real-world data. It will be done in a follow up study, where we address the uncertainties caused by the model and the boundary conditions.

30  "RC: *p.9, l.10: '...200 candidates has a...' → '...have a...'* "
**AR:** Done.

"RC: *p.9, l.16-17: 'in close proximity to the synthetic experiments': OK. Because do not compare to the real observation. Is it not possible to work with real observation? To get match here, would probably have to tune some model parameters. Is*

*this the problematic aspect of working with observed present-day geometry? (seems to be the case when reading the 'Discussion and conclusions' section)* "

**AR:** Yes, the model calibration (mostly for the sliding parameter, the creep parameter and the consideration of lateral drag) also play important roles. This will have to be addressed in a real-world application, but does not influence the results of the synthetic experiments presented here.

"RC: *p.9, l.18: 'the observation': which is also modelled, right?* "

**AR:** Yes, we clarified this in the text.

"RC: *p.9, l.25: results are different: quite trivial. Would reformulate this or simply omit this* "

**AR:** We rephrased this.

"RC:*p.9, l.26: only few with small fitness value: see first comment on this section* "

**AR:** See answer to first comment on this section.

"RC:*p.9, l.27-28: 'need more time to adapt' $\rightarrow$ response time!* "

**AR:** See answer to your first comment (General comments)

"RC: *p.10, l.1: '11.7 to 12.4 km': from this: deduce a retreat of ca. 5 km. Is this the case? Would be really interesting to compare. For this glacier and for many others. Would make story much stronger.* "

**AR:** Please refer to the comments to the real-world applications before.

"RC: *p.10, l.6: 'we were able to show': personally not really convinced at this point unfortunately. See suggestion about incorporating 'real' data (observed past states and observed (length) changes)* "

**AR:** We wrote in the text: "we were able to show, that our method is able to determine the state in $t_0 = 1850$ of the synthetic experiment." and we still think that we could show this very well for the synthetic experiment case.

"RC: *p.10, l.7-8: combining with climatic information. How big is the role of the past climatic information on your results? Does the choice of past conditions (the conditions that you impose) affect the 'best' past states?* "

**AR:** This is an interesting question. The conditions that we impose are not completely arbitrary, as they are taken from HISTALP. But it is a valid question to ask if different past climate conditions (e.g. a climate which is not getting warmer but colder) would affect the accuracy of the method. This question is a perfect example for the usefulness of synthetic experiments: they would allow to address this question (as well as other factors, such as the influence of parameter uncertainty), and could be the subject of a future study. We added a sentence discussing this question in the conclusions.

"RC: *Figure 9+10: number of glaciers mentioned in figure title = 2619 vs. 2621 before. How come?* "

**AR:** In Figure 10, we show the relative error. To calculate the error, we need to divide by the volume of the experiment glacier. Here it happens for two glaciers, that the volume of the experiment was close to zero at some time $t$. That's why these glaciers are not included in this plot. After updating the mass balance related parameters (see General comment2), this issue does not occur any more and the numbers now coincide with each other.

"RC: *p.15, l.1: 'This holds also true': strange formulation. Consider reformulating* "

**AR:** Done.

**Hardware requirements and performance**

"RC:*p.15, l.12: 'influence strongly' $\rightarrow$ 'strongly influences'* "

**AR:** Done.

**Discussion and conclusions**

"RC: *p.16, l.3-4: 'identify errors...introduced by model approximation': OK, but this is not very satisfying for the reader and makes the paper almost purely theoretical...* "

**AR:** The paper presents a new method and thanks to the synthetic experiment world, we are able to determine uncertainties from the method itself. We agree that an actual reconstruction is desirable and plan to do so, but as pointed out above, there are more issues to overcome than fit into one manuscript.

"RC: *p.8-10: synthetic glacier state in 2000. Due to this cannot say something about real glacier state in 1850. Is it not possible to say something about the (length) change during this period and compare this to observations (for the 'best solutions')?* "

**AR:** No, a comparison is not useful. See also comments to real-world applications before.

"RC: *p.13: 'which we don't address here' $\rightarrow$ 'which we do not address here'. A pity...I understand that this is difficult, as explained by the authors (p.16, l.4-7), but question may arise at this point how relevant this story is for the community? Seems to be a good starting point for further research by the group and users of OGGM (p.17, l.26-27: 'framework will be useful'), but more limited for outsiders. Feels more like a 'technical' note/paper now. Here again, I am convinced that by incorporating some 'real data' and/or rethinking of some experiments (see e.g. also suggestion 1 in 'general comments' section): the story would become far more relevant for the glacier (modelling) community.* "

**AR:** We hope that our study is useful for the community (see several comments above). The model and our method are open source and documented.

"RC: *p.16, l.15: 'observed state': really the observed state or a synthetic one?* "

**AR:** We here mean the observed state in general (it can be a synthetic derived one or a real one)

"RC: *p.16, l.16: 'depends on the situation' → 'depends on the specific glacier setting'?* "

**AR:** Done.

5   "RC: *p.17, l.8: 'performs equally good' → 'performs equally well'?* "

**AR:** Done.

"RC: *p.17, l.9: 'In Sect 4.2...': find it strange to have this formulated in such a manner in the conclusion* "

**AR:** We reworded this sentence.

"RC: *p.17, l.12: 'could differ strongly in volume and area': would expect the differences to be relatively small with a trapezium/parabola cross-section, no* "

**AR:** Glaciers with the same length have the same number of grid points with non-zero ice-thickness. At each of these grid points a cross-section (with a parabolic, rectangular or trapezoidal shape) perpendicular to the flowline defines the relation

15   between the ice-thickness and the widths of the glacier at this grid point. Each of these cross-sections can differ, whereas the number of cross-sections is the same. Thus, glaciers with the same length can differ strongly in volume and in area, because they can differ in width (and therefore in ice-thickness).

"RC: *p.17, l.13: 'lead more variability' → 'lead to more variability'* "

20   **AR:** Done.

"RC: *p.17, l.18: 'for some glaciers': indeed, for the ones with a short response time* "

**AR:** Yes, the ones with a low reconstructability measure (indirectly connected to the response time).

**2 Reply to Reviewer #2**

"RC: *The authors present an initialisation strategy for the Open Global Glacier Model (OGGM). The aim of this strategy is twofold. First, the initialisation should produce a best estimate for the glacier extent and geometry at the end of the little ice age (LIA). Second, the spin-up into present day should appropriately reproduce the observed geometry. The latter aim is formulated as an optimisation of an inverse problem. For testing and validation of the initialisation, the authors suggest synthetic experiments for which the past and present geometry is perfectly known. The main target parameter for the optimisation is a so-called 'temperature bias' . The authors show that their strategy allows to constrain the possible parameter space significantly, certainly if the optimisation accounts for geometric information beyond the glacier length. The manuscript is of great interest to the community as it aims at formulating a standard procedure that should provide the initialisation basis for regional or global ice-dynamic simulations of glacier evolution. In this sense, the authors are trying to solve a pressing issue. Moreover, the manuscript is well written and structured. However, I have major concerns on the pursued parameter sampling strategy in light of other a-priori parameter choices concerning ice-dynamics and surface mass balance. Consequently, I fear that the single parameter problem is oversimplifying any real-world application. To address my comments, the authors will certainly have to expand the manuscript to better justify and motivate their decisions. Consequently, I recommend a major revision of the manuscript.*"

**AR:** Thank you for your constructive feedback on our manuscript. We tried to implement most of your comments. However, it is a misunderstanding that "the initialization should produce a best estimate for the glacier extent and geometry at the end of the little ice age (LIA)", since we here focus on the methodological aspect of the reconstruction, and therefore do not present an estimate of the end-of-LIA ice volume in the Alps (among other things). We are aware that we probably did not explain this well enough and tried to improve the manuscript. This choice of experimental setup also has direct implications for the issue of uncertainties from model parameters that the reviewer raises. It is correct that they play an important role for real-world applications and, therefore, we agree that we are oversimplifying the problem (see also our general answer above). By using only one parameter (temperature bias) to create a test case, we are able to separate the uncertainty caused by the method itself from the uncertainty caused by model error, by parameter uncertainty, and by uncertainty of the boundary conditions. We hope we were able to sufficiently clarify our reasoning below and in the manuscript.

**Temperature bias**

"RC: *For the synthetic experiment, you prescribe a 'random climate scenario'(p. 5,line 26). I think that this term refers to a random permutation of the climatic forcing around the year 1850 (within a 30 year period). Is that right?*"

**AR:** Yes, the random climate scenario is explained on p.4, line 6-9. We shuffle the years (using a 31 year period around 1850) infinitely.

"RC: *I assume that the climatic forcing is taken from HISTALP. You then add a temperature bias of -1K to this climatic forcing. This bias is not well motivated. Why is it necessary?*"

**AR:** The main motivation is to create a synthetic environment which is realistic, i.e. close to what we would expect from the real word. When created with such a climate, our synthetic 1850 states are such that when evolving under the 1850-2000 climate they reach a state which (on average) is that from the Alps in 2000. To justify the choice of $\beta$=-1 K, we also test different values. The results are shown in Fig 3 (see p. 18). This figure shows that applying positive or small negative temperature biases to the synthetic experiments results in large area differences to the RGI in 2000, and the total glacierized area in 1850 is also too small. The sample size is reduced, because less glaciers fulfill the area threshold criteria of 0.01 km$^2$. Negative temperature biases that are too large also reduce the sample size, because some runs fail ("Glacier exceeds boundary", this means the glacier would get larger than the underlying grid). The experiments with a temperature bias of -1K, -1.25K or -1.5K perform best regarding the area difference to the RGI in 2000. But only the experiment with the temperature bias of -1K performs well regarding the estimation in 1850 of Zemp et al. (2006), whereby it needs to be taken into account that the dots only represent a subset (the small glaciers in 2000 are missing) of the glaciers considered in (Zemp et al., 2006). We added this results to Appendix A.

Additionally,we illustrate the process of the generation of the synthetic experiment again with the example of the Hintereisferner (see Fig. 3). Under climate conditions similar to the climate around 1850, today's glacier would still lose mass. If we do not apply a negative temperature bias to the random climate run, we would create a synthetic experiment with a glacier in 1850 that is smaller than today's glacier.

"RC: *I thought that HISTALP provide you with perfect climate conditions*"

**AR:** Yes, they do. Note that the temperature bias is only applied to create the true initial states (in which case the bias is fixed but not used by the reconstruction method since it should be unknown), and then by the reconstruction method to generate the potential candidates (in which case many biases are used and tested for). For the past climate runs from 1850 to 2000, we do not change the temperature bias, as HISTALP provide us "perfect" climate conditions for this time period. We add in the manuscript the information that no temperature bias is applied for the past climate runs (see Sect. 2.4.3 Evaluation) to avoid further confusion.

"RC: *Please provide a firm motivation for this bias, why it varies from glacier to glacier and consequently has to be inferred*"

**AR:** The starting point of our method is always the present state, as this is the only state we know. Provided with the assumption that past glaciers were different than today, we need a way to create physically consistent candidates. Glaciers respond differently to changes in climate and thus the required temperature biases to create these candidate are different for each glacier, too. As we do not know the size of the glaciers in 1850, we also do not know what temperature bias is necessary to create a glacier with a given size in 1850. This is the reason why we create a large set of physically plausible glacier candidates, which

[Figure]

**Figure 3.** Generation of the synthetic experiment for the Hintereisferner. The dashed grey line marks the volume of today's glacier, the blue line the evolution without a temperature bias during the random climate run and the orange line using a temperature bias of -1 K. **a:** Generation of the synthetic experiment in 1850. These runs are forced with a random climate scenario (31 years around 1850). The glacier state at the end of this run (t=600) is the synthetic experiment in 1850.**b:** This state serves as initial condition for the past climate run (forced by HISTALP, no temperature bias) to create the synthetic experiment in 2000.

we need to evaluate with the forward run from 1850-2000. The range of biases we test in our study is not entirely random, as we assume that the glaciers are likely to be as big (or bigger) than the present-day geometry. But the method does not depend on this range: it can be increased at the cost of more computations (as we would have more candidates to test for). In the text, we motivated the use and the necessity for the temperature bias in Sec. 2.4.1 "Generation of glacier states".

**5  A-priori OGGM calibration vs. parameter sampling**

"RC: *Initially, the first question which came into my mind was why you did not enlarge the ensemble by including other uncertain parameters, as for instance the rate factor, basal friction or parameters linked to the SMB model. So ultimately this question links to the comment above. In other words, how do you ensure that the other parameters are well constrained.*"

**AR:** Including other uncertain parameters is of course possible. Adding e.g. a bias for the precipitation during the random climate run would be an option, too. We think that the variations we could obtain by varying "only" the temperature bias, the permutation and the time point (during the selection of the candidates) are sufficient and we could well demonstrate the functionality of our method. For the moment we ensure that other uncertain parameters are well constrained, because we used a synthetic environment: there is no parameter uncertainty influencing the results presented here. We added more information about the choice of these parameters in the text and explain how other parameter choices might influence the uncertainty in real-world applications in the discussion.

"RC: *I understand that you present a synthetic setup but with regard to any real-world application this issue is very important. Even if there was an a-priori calibration of a combination of the rate factor and the melting parameters, other combinations might also produce plausible glacier geometries at present. Yet the exact choice will affect the past glacier geometry.*"

**AR:** Yes, we agree. The exact choice of these parameters will affect real-world applications. Parameters linked to the SMB model are validated trough a cross-validation. The most important uncertain parameters of OGGM that will influence the past geometry as well as the response time of the model are the sliding parameter $f_s$, the creep parameter $A$, and the consideration of lateral drag. Maussion et al. (2019) analyzed the model sensitivity to these parameters. But again, as long as we consider synthetic experiments, this does not affect the results. For real-world application, the calibration of these parameters and the quantification of their influence on our results has to be part of the reconstruction procedure. The complexity of this issue is why we decided (and believe that it is necessary) to first realise this synthetic study.

"RC: *In many other glaciological applications, basal friction or the rate factor are the central unknowns that are calibrated during the model initialisation. I therefore think that it is inevitable that you include a section on the OGGM procedure for the calibration/choice of these other parameters. On this basis, you should motivate your parameter choice for your ensemble generation. Dependent on how the other OGGM parameters are calibrated, it might well be necessary to include further parameters in the ensemble generation*"

**AR:** We agree, and are well aware that parameter uncertainty will play an important role in real-world applications. The purpose of this study, however, is to explicitly remove parameter uncertainty and still demonstrate that, even in this controlled setting, the problem of model initialisation is non-trivial and non-unique. We hope that the changes in the introduction and method section of our revised manuscript now make a better case at explaining this point.

**Climatic forcing**

"RC: *Throughout the manuscript, I sensed some redundancy because I did not find any discussion of the 'acceptable ensemble states' in terms of the initial parameter choice. To put it bluntly, were you able to infer the -1K temperature bias prescribed in the synthetic experiment from your data assimilation? Otherwise, the specific climatic permutation might have had a significant influence?*"

**AR:** The permutation is necessary to create the random climate time series that are 600 years long (see also the point after the next point). We were able to reproduce the -1 K temperature bias from the synthetic experiment. Figure 4 shows a histogram of the temperature bias of all median states (best candidates). The bias of the majority lies around -1 K. Values that differ more from -1 K, can be explained by the uncertainty of the median state. Especially in cases where most tested candidates are acceptable, the median is only a representative of the large set of acceptable glacier states. The aim of this study is to determine a past glacier geometry, not a temperature bias. The temperature bias is (only) the tool, which creates the desired geometry candidates. For this reason we decided to do the uncertainty analysis based on the glacier geometry and not based on the temperature bias.

[Figure]

**Figure 4.** Histogram of the temperature bias from median states.

"RC: *In short, please justify your choice to exclusively focus on climatic quantities in the optimisation. From my experience, you should include other parameters in this optimisation.*"

**AR:** Our method searches the climate conditions (similar to the ones around 1850) under which the today's glaciers would grow to their "true" size in 1850. Dynamical parameters of the model would have to be calibrated before the initialization
5 method (see also comments to parameter calibration).

"RC: *I wonder about the necessity to permute the initial climate time series during the initialisation. Do you really attain distinctly different glacier geometries in 1850 that you could not generate by only changing β. Please re-assess the dual sampling of climatic variables/input in the ensemble generation*"

10 **AR:** The infinite shuffling does not only create more variations, but is also necessary to create a time series that is long enough to reach the stagnation period. Without the random shuffling, we would not be able to create the 600 year long time series which we use for the generation. Of course, we could use the same permutation for all runs and only varying β, but there is no reason to prefer only one out of the huge set of possible permutations. For this reason we decided to use for each different β, which we tested, a different way to shuffle the 31 years around 1850. We added this argument in the manuscript (see Sec. 2.4.1)

"RC: *The past-to-present volume evolution of Guslaferner of all ensemble members is shown in Fig. 3. It surprised me that all ensemble members readily converge to a very similar present-day value. Certainly if you consider the large range of β-values used for the initialisation.*"

**AR:** The rapid convergence was caused by the wrong parameter set used for the SMB calibration (see also comment to your question about the calibration of the model). We repeated all runs with the correctly calibrated mass-balance model. The convergence is delayed now.

**Detailed comments**

5  "RC: *on page 8 lines 10- 11, my attention was then drawn to the fact that all forward runs are conducted with 'exactly the same climate time series' and use the 'same parameter set'. Does this include $\beta$?*"

**AR:** Yes, this include the temperature bias $\beta$=0. The past climate runs (from 1850-2000) use the HISTALP time series, which provide us perfect climate conditions (as mentioned earlier). There is no need to change the temperature bias for these runs. The climate, as well as all other parameters are the same for the forwards runs starting in 1850. Only the initial condition

10  (surface elevation and widths along the glacier flowline) is different.

"RC: *If $\beta$ was set to zero after the initialisation, I would be highly concerned about the abrupt climatic shift you introduce when switching from the initialisation to the forward experiment. A reason for the latter case is the overall quick convergence during the forward simulations*"

15  **AR:** The past climate runs are performed with temperature bias $\beta = 0$, but the generation of the candidates is distinct from the initialization process and thus we do not see a major issue that arises due to an abrupt climatic shift. From the selected states we only take the surface elevation and the widths along the flowline and we start a completely new run.

"RC: *What climatic forcing is used for the a-priori OGGM calibration. Is it HISTALP. Is the temperature bias $\beta$ included?*

20  *Please clarify.*"

**AR:** The mass balance model calibration is done with HISTALP[1]. We added this information in the manuscript in Sect. 2.1. The temperature bias (which is an artificial offset that we use to generate glacier candidates) is set to zero for the calibration.

"RC: *p.1, line 20 - Sentence is difficult to understand. Reformulate.*"

25  **AR:** We rephrased the sentence.

"RC: *p. 2, line 17 - 'Despite of the importance ...' –> 'Despite the importance ...'*"

**AR:** Done.

30  "RC: *p. 2, line 28 - Introduce a comma (,) before which. Please check throughout the document.*"

**AR:** Done.

"RC: *p. 4 ,line 1 - '... glacier's length ...' –> '... glacier length ...'. Please avoid the 's genitive throughout the document.*"
* * *
[1]See our cross-validation monitoring website if you are interested in the results: https://cluster.klima.uni-bremen.de/~github/crossval

**AR:** Done.

"RC: *p. 5 ,line 5 - Please omit the bedrock difference in Equation (4). As the bedrock does not change during your initialisation this term is zero*"

5   **AR:** Done.

"RC: *p. 5 ,line 22 - Remove comma.*"
**AR:** Done.

10   "RC: *p. 5 line 28 - Delete 'this' at the beginning of the line.*"
**AR:** We corrected the sentence.

"RC: *p.6 line11-p.7 line 5 - The initialisation ensemble is formed by the variation of two parameter: $\beta$ and a permutation of the 30-yr climate forcing time series. For both quantities it remains unclear how many ensemble members are created. As*

15   *you reduced the ensemble later on to 200 members based on an equal spacing argument, I would assume a sufficiently dense sampling. Anyhow, please provide some numbers.*"
**AR:** The number of ensemble members differs from glacier to glacier. This depends on $t_s tag$ and on the number of successfully completed random climate runs $n_r$ (number of grey lines shown in Fig. 2). We stored yearly glacier volumes and thus the ensemble size is $n_r(600 - t_{stag})$. In the case of the Guslarferner $t_{stag} = 282$ and $n_r = 140$ result in $\sim 44500$ members.

20   The sampling is sufficiently dense in all cases. We add further information in the manuscript in Sect. 2.4.2 Identification of candidates.

"RC: *p.6 line 13 - It remains vague how the permutation is done. You permute the climatic forcing per year, month, etc.*"
**AR:** We shuffle the years, but use monthly climate data. The order of the month are not changed. We made this more clear in

25   the text.

"RC: *p.9 line 6 - Please provide values for glacier area and estimates for mean ice thickness for Guslarferner and Hintereisferner at present. Hintereisferner covers a much larger area and is probably much thicker. These values are informative and they are difficult to infer from Figs. 3 and 5.*"

30   **AR:** We add this information to the captions of Fig. 3 and 5 in the manuscript. Additionally we improved the labelling of the x-axis of these figures. Please keep in mind, that the values of the synthetic experiment can differ to the reality.

"RC: *p. 9, line 10 - 'has' –> 'have'*"
**AR:** Done.

35

"RC: *p. 10, line 2 - 'not' –> 'no'*"

**AR:** Done.

"RC: *Figure 2 There is a discrepancy between the range of temperature biases given in the caption [-2.65, 2.95] and shown*
*in panel (a) [-3.0, 1.95]. Moreover, I would try to remove some redundant information. For panel (b), limit the graph to the*
*stagnation period. For panel (c), do show the initialisation period with the coloured points but extend the figure by the forward*
*simulation 1850-2000 as in Fig3c.*"

**AR:** We corrected the values for the temperature range in the figure and limit the graph in Figure 2b and 2c to the stagna-
tion period only. We decided against the extension of the simulation 1850-2000, because the Figure becomes too confusing
otherwise.

**References**

Maussion, F., Butenko, A., Champollion, N., Dusch, M., Eis, J., Fourteau, K., Gregor, P., Jarosch, A. H., Landmann, J., Oesterle, F., Recinos, B., Rothenpieler, T., Vlug, A., Wild, C. T., and Marzeion, B.: The Open Global Glacier Model (OGGM) v1.1, Geoscientific Model Development, 12, 909–931, https://doi.org/10.5194/gmd-12-909-2019, https://www.geosci-model-dev.net/12/909/2019/, 2019.

5 Oerlemans, J.: Climate Sensitivity of Franz Josef Glacier, New Zealand, as Revealed by Numerical Modeling, Arctic and Alpine Research, 29, 233 – 239, https://doi.org/10.80/00040851.1997.12003238, http://www.jstor.org/stable/1552052, 1997.

Zemp, M., Haeberli, W., Hoelzle, M., and Paul, F.: Alpine glaciers to disappear within decades?, Geophysical Research Letters, 33, https://doi.org/10.1029/2006GL026319, https://agupubs.onlinelibrary.wiley.com/doi/abs/10.1029/2006GL026319, 2006.

---

## Author Comment (AC2) · 27 Aug 2019

Please find our comments to both referees in the supplement PDF of AC1.